# Quantifying the direct and indirect protection provided by insecticide treated bed nets against malaria

H. Juliette T. Unwin [1] ✉, Ellie Sherrard-Smith [1], Thomas S. Churcher [1] & Azra C. Ghani[1]

Long lasting insecticidal nets (LLINs) provide both direct and indirect protection against malaria. As pyrethroid resistance evolves in mosquito vectors, it will be useful to understand how the specific benefits LLINs afford individuals and communities may be affected. Here we use modelling to show that there is no minimum LLIN usage needed for users and non-users to benefit from community protection. Modelling results also indicate that pyrethroid resistance in local mosquitoes will likely diminish the direct and indirect benefits from insecticides, leaving the barrier effects intact, but LLINs are still expected to provide enhanced benefit over untreated nets even at high levels of pyrethroid resistance.

Long lasting insecticidal nets (LLINs) have been attributed with averting 663 (542–753 credible interval) million clinical cases (68% of malaria cases) globally across 2000 to 2015[1]. In recognition of this impact, most malaria-endemic countries in sub-Saharan Africa distribute LLINs universally to communities through mass campaigns operating approximately every 3-years[2,3]. Adherence to net usage is variable among communities but generally good when access is good[4]. Variable access and usage can be partially explained by uncomfortably high humidity and temperatures in some areas making sleeping beneath nets challenging[5], an absence of perceived risk, misinformation on LLIN utilization[6,7], and coverage gaps in distribution campaigns[5,7,8]. Recently, school-based and health centre 'top-up' campaigns are trying to address such coverage gaps[9].

LLINs contain insecticides which kill mosquito vectors so that where net use is not universal, some community protection is nevertheless potentially provided to everyone. The challenge to deliver LLINs universally and the emergence of mosquitoes able to survive exposure to pyrethroid insecticide—historically, the principle active ingredient for LLINs—has led malaria researchers to question the protection offered by insecticide[10,11]. Others strongly advocate that the killing effect of LLINs is integral to their continual protective benefit, more so than universal coverage of a community[12]. As shown in previous research such as Hawley et al.[13] and Killeen et al.[14,15], LLINs offer different benefits for users and non-users within a community:

net-users receive personal protection while both users and non-users benefit from indirect protection. This is due to the reduced numbers of mosquitoes, and reduced proportion of these mosquitoes which are infectious (due to higher mosquito mortality and lower human infectious reservoir)[15–17]. This logic, shown empirically[13,18] and theoretically using mathematical models[15,17], formed the basis for the adoption of universal coverage with LLINs as a global policy by the World Health Organisation (WHO)[12,19]. It is possible to quantify the direct and indirect protection offered from LLINs to both users and non-users further into benefits afforded by the barrier distinctly to benefits from the insecticide. Doing so can contribute to the debate on the use of untreated nets[10,11] and can inform potential loss in impact due to pyrethroid resistance in local mosquito vectors. The four mechanisms determining the overall level of protection are summarised in Table 1.

Most studies measure the overall efficacy of LLINs through cluster randomised control trials[13,20–22]. However, we consider direct protection to the net-user from two components: direct protection attributed to the barrier and direct protection attributed to the effects from the insecticide that kills or deters mosquitoes from biting the protected individual. Here we separate the barrier from the actions of the insecticide (i.e. the barrier effect but not the insecticidal actions are seen in untreated nets). The whole community benefits indirectly from the barrier and the insecticide given lower burden of infection across the community. Previously the contribution from both these terms has

[1]MRC Centre for Global Infectious Disease Analysis, Department of Infectious Disease Epidemiology, Faculty of Medicine, Imperial College London, London, UK. ✉e-mail: h.unwin@imperial.ac.uk

**Table 1 | Definition of the four different mechanisms determining the overall level of ITN protection and the equations used for their estimation**

| | Direct<br>net users | Indirect (community effect)<br>net users and non-users |
|---|---|---|
| Barrier | (a) Reduction in EIR due to reduced mosquito blood-feeding rate. | (b) Change in EIR for net users and non-users caused by lower human prevalence resulting from (a). Non-users may experience more bites from mosquitoes repelled away from net users |
| | Calculated as control − $A$ | Calculated as $A − B$ |
| Additional protection from insecticide | (c) Further reduction in EIR caused by insecticide killing and deterrence of blood-seeking mosquitoes | (d) Further reduction in EIR due to higher mosquito mortality, lower mosquito abundance and lower human prevalence resulting from (c) |
| | Calculated as $A − C$ | Calculated as $B − D − (A − C)$ |

The four mechanisms (defined a–d) are described according to their impact on transmission. Direct benefits occur to those using ITNs, whereas indirect benefits act upon everyone in a community where ITNs are in use. The value of each are estimated from a transmission dynamics mathematical model, with the different scenarios (italic upper case letters) refer to model run with different assumptions presented in Fig. 2.

been referred to as the community effect. One early study providing evidence of the community effect comes from a cluster randomised trial of insecticide treated nets (ITN) conducted in western Kenya[13]. In this trial a clear gradient of impact was observed in the control areas in which ITNs were not distributed but that were close neighbours of areas in which ITNs were distributed, with a reduction across several different malaria-related outcomes including malaria prevalence and parasitaemia. ITN usage in the intervention areas was observed to be around 70%[23]. Other early trials enable the estimation of different components of protection e.g. direct protection from barrier + insecticide was quantified in[13,24,25], direct protection from the barrier in[26], and direct protection from insecticide in[27] (Table 1).

Inferences can be drawn from other early studies that empirically tested malaria burden in users and non-users and these tend to show intuitively that burden is lower in the cohorts using nets and some parallel, but smaller reductions are often seen in non-user cohorts[13,28,29]. More recently, Killeen[12,15] used models to argue that personal protection is only a minor fraction of the overall effect. Evidence against a community effect sometimes cite a study conducted in The Gambia[30] because malaria prevalence was higher among non-users living within village clusters of people using nets than within villages without nets. However, study design may explain this because mosquito subpopulations in intervention villages and non-intervention villages may have mixed[31].

Evidence is now building that pyrethroid resistance in mosquito-vector populations is leading to diminished protection from LLINs that use this insecticide as the active ingredient[32–35]. In parallel, evidence is emerging that different net brands may be more robust than others, potentially offering longer direct benefits to net users than others[36]. Both the barrier and the insecticide of the net offer direct and indirect protection to both users and non-users. However, these may be differently impacted by the presence of pyrethroid resistance in local mosquito populations or the integrity of the netting material. In this context, it could be useful to quantify the different types of benefit offered by LLINs so that we can start to consider how to mitigate against lost personal or community protection by coupling nets falling short with alternative interventions or focusing research and development efforts on enhancing particular benefits[12].

It would be unethical now to test these distinctions empirically because it would require leaving a cohort of people without nets and therefore exposed to potential transmission risks. In addition, it is difficult to isolate the four comments of protection offered to users and non-users of ITNs in a community, nor show how these might change with varying population usage of ITNs or levels of pyrethroid resistance. In this manuscript, we compared the difference in prevalence between users and non-users of LLINs in a mechanistic transmission model of falciparum malaria[37–40]. To provide some confidence in the model we statistically analyse Demographic Health Survey (DHS) data[41] to explore whether qualitative predictions made by the model

are supported by epidemiological evidence. These data are produced every few years from sentinel settings and summarise, among many other aspects of health: (i) the proportion of people in the assessed cluster using nets, (ii) those having access to nets, and (iii) the prevalence of malaria parasite infections detected by rapid diagnostic test at the individual level. We then use a transmission model to tease apart the direct benefit of LLINs from the mass community effect and investigate what happens as mosquitoes show increasing resistance to pyrethroid insecticide. We discuss these findings in the context of previous work quantifying personal protection and the community effect.

## Results
### Illustration of model outputs
We illustrate the process we take to decouple estimates of prevalence in users and non-users of mosquito nets in Fig. 1a for both treated and untreated nets (see Supplementary Fig. 1 for net parameterisation insecticidal impact decay through time assuming a 3-year net distribution cycle). For a population of 100,000 people, we simulate LLINs being used by 50% of the population at random from year 1 and show the all-age prevalence across the whole community falls from 60% to ~40% by year 2 for treated nets and 56% for untreated nets. In the example, it is clear the assumptions of the model indicate that more protection is afforded to net users—reducing all-age prevalence of insecticide-treated net users to 33%—whilst there is less reduction in the non-user cohort for the treated net scenario (all-age prevalence of non-users is reduced to about 47%). The reduction in prevalence predicted by the model varies depending on the initial entomological inoculation rate (EIR, the mean number of infectious mosquito bites received per person per year) simulated, but some protection is offered to non-users by having any nets in the community (Fig. 1b). The model is used to project how malaria prevalence may differ between users and non-users for different levels of malaria endemicity and LLIN use. Results predict that the absolute difference in malaria prevalence is greatest in areas with intermediate levels of malaria, peaking at around a 16% difference at 50% rapid diagnostic test (RDT) prevalence (Fig. 1c). Interestingly, the model projects the absolute difference in malaria between users and non-users to be consistent irrespective of LLIN usage in the population (Fig. 1c).

### Comparison of model outputs to DHS data
Figure 1d shows the prevalence between users and non-users for children 6–59 months of age from the DHS data. For illustrative purposes data are binned into 10% prevalence bands according to overall prevalence observed in the cluster. Consistent with the model output we see there is only a modest difference between prevalence in users and non-users in each band, peaking in clusters where approximately half the cluster were positive for malaria. At very low (<20%) or very high (>80%) usages the difference between users and non-users is relatively

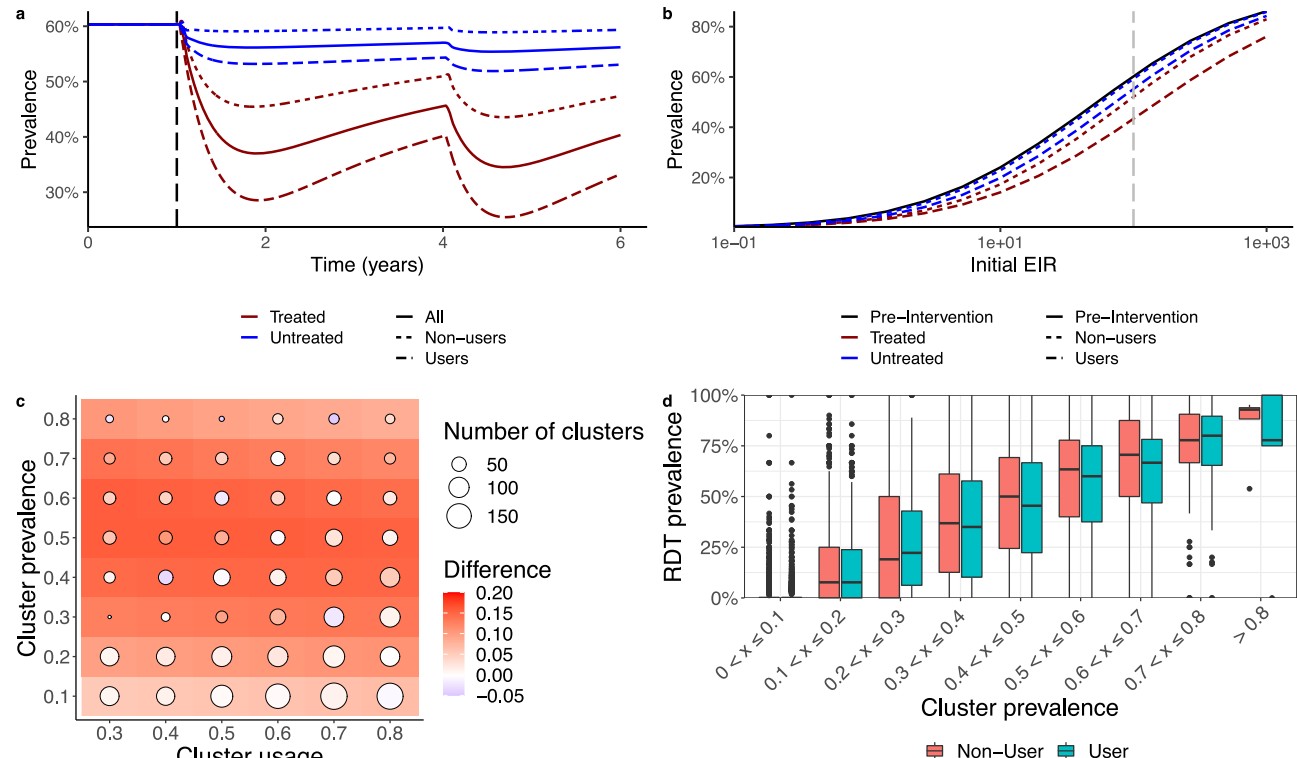

**Fig. 1 | Protected impact on malaria prevalence of standard pyrethroid long-lasting insecticidal nets (LLINs) for net users and non-users. a** All-age malaria slide prevalence for a perennial setting with an initial entomological inoculation rate (EIR) of 100 bites per person per year. At year 1 (indicated by the vertical dashed black line), in this example, 50% of the population switch to using LLINs. **b** The dependency of all-age malaria parasite prevalence on the annual initial EIR for the baseline pre-intervention scenario (black solid line), the LLIN users (blue dashed line), and non-users (red line). The initial EIR of 100, which is the simulation shown in A, is indicated by the vertical dashed line. **c** The absolute difference in prevalence between LLIN users and non-users aged 6–59-months. The coloured tiles show the difference for our model estimates and the coloured points show the difference from the DHS survey (size represents number of data points). **d** RDT prevalence from DHS surveys for users (blue) and non-users (red) for different cluster prevalences. 4138 clusters have been used for this figure (one value for users and non-users), with the centre of the box and whisker plots showing the median, outside lines showing the first and third quartiles, and the whiskers indicating 1.5 times the interquartile range. Dots indicate outliers.

inconsistent, often with higher malaria observed in net users (Supplementary Fig. 2). It is unclear whether this is due to a negative association between malaria risk and net use (i.e., people more at risk are more likely to use a net) or reflecting the stochastic nature of estimating prevalence when the number of people in the category are low. We note that in some extreme clusters all users or non-users may have malaria so individual clusters may have a difference in prevalence between −1 and 1. Figure 1c breaks cluster-level data down further, categorised by prevalence and the level of net usage among these children. Results are noisy, but broadly match those observed in the model.

To statistically explore whether our model projections reflect the empirical evidence from the DHS data we conducted mixed-effect logistic regression. Clusters were grouped into low (0–33%), medium (33–66%) and high (66–100%) malaria prevalence and low (20–50%) and high (50–80%) net usage. We see that only the medium malaria endemicity group has a significant difference between users and non-users (median of 3.2% higher, $p = 0.017$). Supplementary Fig. 3 shows the raw data binned in these groups whilst Supplementary Fig. 4 shows the modelled predictions for the difference between users and non-users. Neither year of survey or cluster net usage significantly improves model predictions of the difference in prevalence between non-users and users. This supports the hypothesis generated by the mechanistic model that the absolute difference in prevalence between users and non-users is not influenced by community levels of net usage but rather is associated with the local endemicity of a setting. The interaction term between prevalence and usage categories is insignificant ($p = 0.197$ and $0.499$), suggesting the difference in malaria prevalence between users and non-users is also consistent. Overall, there is less of a difference in the field data between users and non-users for a given prevalence compared to the model suggesting current model structural assumptions may be exaggerating the difference. Analyses across all net usage levels is shown in the supporting information.

Similarly, we did not detect a signal using linear regression from the DHS data aggregated to the first administration level that the difference between users and non-users changed with the level of pyrethroid resistance ($p = 0.654$). Using the mechanistic model, the impact of resistance on the difference between users and non-users is estimated to be relatively modest (Supplementary Fig. 5), which is supported by the lack of a clear trend in the DHS data.

## Model predicted direct benefit of barrier
The mechanistic model suggests that the relative magnitude of the direct and community effect varies according to the level of disease endemicity and usage of nets. In the first instance, we assume that no mosquitoes are resistant to the insecticide used on the nets. In addition to a control where nobody in the community is given any nets, we identify 4 scenarios that describe the different types of protection offered by untreated nets or LLINs to users and non-users (Fig. 2). Results for these different types of protection are shown as their ability to reduce EIR in a hypothetical non-seasonal setting. The exact values should be treated with caution as they reliant on uncertain model assumptions, though the relative magnitude of the different types of protection are likely to be more robust as they are less sensitive to these assumptions.

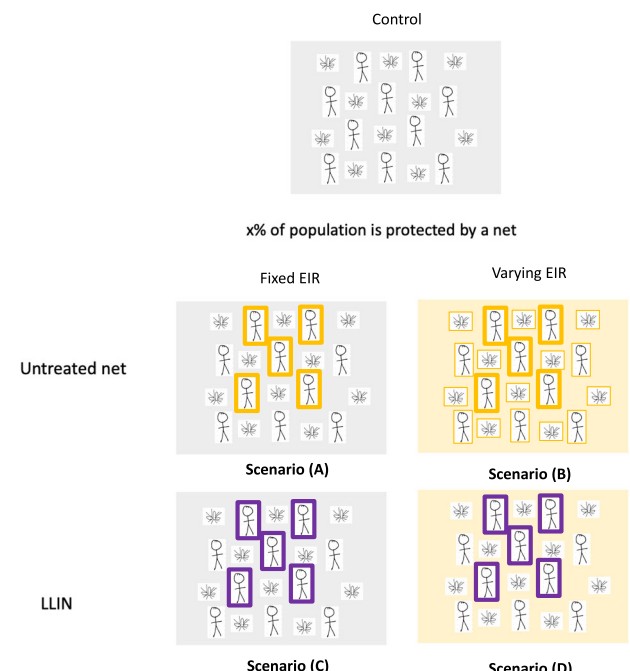

**Fig. 2 | A schematic of control and 5 different theoretical scenarios.** Top panel: Nobody uses a net in the control scenario, and this is used as the counterfactual to compare the other scenarios against. Bottom panels: Scenario (A) illustrates direct protection from the barrier to a proportion of the population, in the analysis we consider an individual (1/100,000),10%, 50% or 80% net use. If x is illustrated, 50% of people consistently used an untreated net (orange rectangle) for a fixed entomological inoculation rate (EIR) (i.e. the entomological inoculation rate, EIR, remains constant irrespective of the net use, grey box). Scenario (B) illustrates direct + indirect protection from the barrier to a percentage of the population. This time we illustrate that 50% of people are protected with an untreated net (orange rectangle) for a varying EIR (i.e. EIR varies over time according to the net use as estimated by the mathematical model, yellow box). Similarly, scenario (C) illustrates direct protection from the barrier and insecticide to a percentage of the population via an insecticide treated net (purple rectangle). Scenario (D) illustrates direct + indirect protection from the barrier and insecticide to a percentage of the population.

First, we consider how the direct protection offered by a barrier reduces the EIR at different levels of mosquito net usage. In scenario (A) there is no insecticide on the net, so the protection provided by the barrier is assumed to be the same as to users of untreated nets (Fig. 2).

When one user in the population is protected, the direct impact from the barrier to that one user is a relative reduction in EIR of 29% [95% uncertainty interval (UI): 13–53%] (Fig. 3a) from the original EIR of 100 infectious bites per person per year. Under the assumptions of the model, we express our findings as reductions in EIR relative to the starting EIR of 100 infectious bites per person per year so here, the effect size for any individual user would be a 29% relative reduction in infectious bites per year.

The level of EIR reduction from the barrier to a net-user does vary with usage level. As the usage increases, the direct protection from the barrier to those using an untreated net progressively reduces, when as many as 80% of the population are using nets, the direct protection for an individual from the barrier is a relative reduction in EIR of 14% [95% UI: 6–31%] (Fig. 3a). This is due to the repellence action of the untreated barrier, which causes the number of mosquito bites on users to increase as more people use nets. When there is only one net user in the population, they will be highly unlikely to receive any bites from mosquitoes which have attempted to feed elsewhere. As usage in the population increases, the overall number of mosquito feeding attempts will also increase, as those mosquitoes that previously tried

to bite net users but were repelled search for a blood-meal elsewhere. These biting attempts are evenly distributed in the population so fall on both net users and non-users. As protection from untreated nets is partial, the EIR on users will therefore increase, reducing the direct benefit, even though the model assumes that some mosquitoes die (at their natural death rate) during the time it takes for a mosquito repelled by a net to searching for an alternative blood-meal (i.e., the probability of feeding and surviving decreases each feeding attempt as time passes). Non-users will also see an increase in the number of feeding attempts, though due to our definition of what constitutes a direct benefit these potential increases in biting are incorporated in the indirect section of Fig. 3.

We can also summarise the direct benefit of the barrier to the combined community (users and non-users). When one person in the community uses a net, the relative reduction in EIR at the community level is minimal (0% [95% UI: 0–1%]). However, when 80% of the community are using nets, there is a relative reduction in EIR of 11% [95% UI: 5–25%] to the community.

### Model predicted indirect benefit of barrier

The indirect benefit of the barrier is the protection provided to the community through lower malaria transmission due to people using untreated nets. In Fig. 3 this protection is shown for users, non-users and the community in general according to the level of net usage. Using the model, we predict that indirect benefits from only one net in the community (scenario (A)–(B)) are minimal but that the magnitude of this type of protection increases with usage. When 80% of the community are using nets, we estimate a relative reduction in EIR of 24% [95% UI: 5–41%] for users, 12% [95% UI: −3–40%] for non-users and 22% [95% UI: 4–41%] for the community (Fig. 3b). The impact of increasing untreated net usage on the indirect benefit to non-users is a trade-off between potentially receiving more mosquito bites (caused by mosquitoes being repelled from net users) vs the reduction in the probability that those bites are infectious (caused by fewer people in the community having malaria and the extended foraging time required to successfully blood feed). Our model suggests that for most scenarios, as usage increases, the indirect protection provided by the barrier effect increases (i.e. the reduction in the sporozoite rate has a greater impact on the EIR of non-users than the increase in their biting rate). However, there are some rare scenarios explored in the sensitivity analysis when the indirect benefit to non-users may be negative (as seen by some negative values in Fig. 3b). Greater indirect benefits are seen in users rather than non-users because, as net usage increases, net-users experience less of an increase in mosquito bites caused by mosquitoes being dissuaded from feeding on users (conversely, non-users will receive more bites as net use increases as more mosquitoes will be repelled from users).

Supplementary Fig. 6b illustrates the combined direct and indirect benefit of an untreated net (the barrier impact alone). It indicates that the increase in protection from the indirect benefit of the barrier at higher usage levels to users outweighs the decrease in direct protection from the barrier at higher usages. Therefore, the total protection (direct + indirect) from a barrier for users increases with usage: it is a reduction in EIR of 29% [95% UI: 12–53%] for one individual using a net, which increases to 42% [95% UI: 11–71%] when 80% of the population use a net. This corresponds to no meaningful reduction in EIR when just one individual uses a net, to a 12% [95% UI: −3–40%] relative reduction in EIR for non-users (when 80% of the population are using nets). At the community level, again there is no meaningful reduction when just one person uses a net but the relative reduction when 80% of the population use a net is 36% [95% UI: 8–65%].

### Model predicted direct benefit from insecticide

The addition of insecticide on LLINs increases the mosquito mortality and reduces the probability of repeating a feeding attempt. We isolate

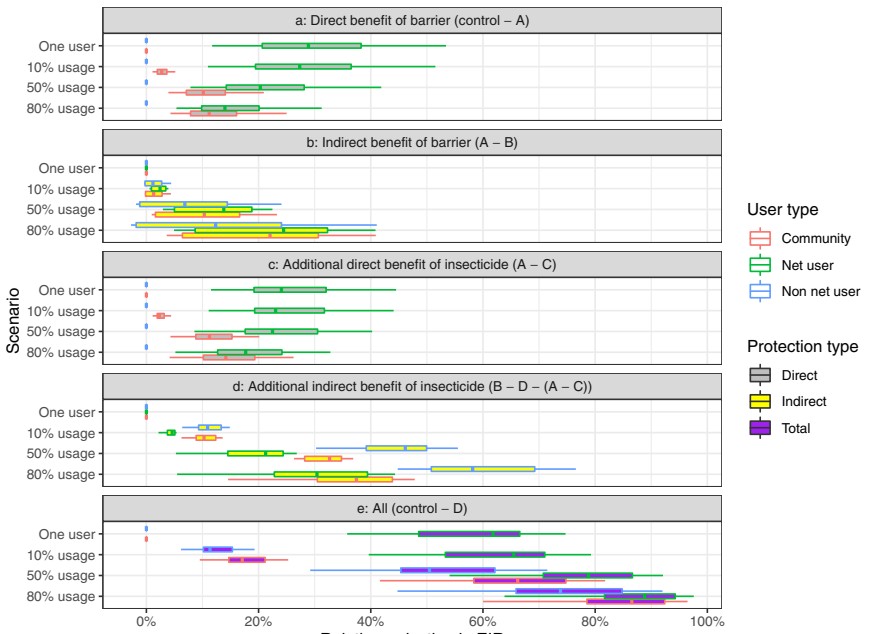

**Fig. 3 | Relative reduction in entomological inoculation rate (EIR) from direct and mass community (indirect) protection offered by mosquito nets for a pre-intervention EIR of 100.** The reduction in EIR is calculated relative to a control scenario, where nobody in the population is given a net, for the five scenarios detailed in the methods section. Scenarios (A–D) (see Fig. 2) are repeated for an individual using a net and 10%, 50% and 80% of the population using nets. The five subplots (**a**–**e**) show results for a different type of protection. The reduction for users is shown in green, non-users in blue and the whole community in red. Direct reductions in EIR are filled in grey, indirect reductions in yellow and total reductions in purple. Box-plots show the range of uncertainty generated by the sensitivity analyses. Fifteen samples were used to generate each box. The centre of the box and whisker plots shows the median, outside lines showing the first and third quartiles, and the whiskers indicate 1.5 times the interquartile range. Dots indicate outliers.

the additional benefit of this insecticide from the direct benefit of the barrier by subtracting the impact from scenario (C) from the respective scenario (A) (Figs. 2 and 3c). The direct benefit of insecticide does not impact non-users (blue bars) but reduces for users (green bars) with increasing usage (the relative reduction in EIR for that individual is 24% [95% UI: 12–44%] when only one individual has a LLIN and 18% [95% UI: 6–32%] when 80% of the population use LLINs). The same pattern occurs at the community level with the direct benefit of insecticide as the direct benefit of the barrier with negligible relative reduction in EIR for one individual using a LLIN but 14% [95% UI: 5–26%] when 80% of people are using them. This is again caused by the deterrent/repellent nature of the insecticide which dissuades mosquitoes from entering houses with LLINs and increases the chance they feed elsewhere. As usage goes up the number of mosquitoes attempting to enter houses increases (as less bites are successful) and the direct benefit of the insecticide to users diminishes, though the relationship with usage is less pronounced (than in 3a) due to the killing actions of LLINs (i.e., fewer mosquitoes are repelled as more are killed).

**Model predicted indirect benefit from insecticide**
Finally, we see that the indirect additional benefit of insecticide increases with usage for both user and non-users. The indirect impact of insecticide mostly increases the reduction in EIR, as expected, due to the killing action of the insecticide. This is minimal when only one user in the population has a net (reduction in EIR for both users, non-users and community is 0% [95% UI: 0–1%]) (Fig. 3d), but further reduces the EIR by 30% [95% UI: 7–44%] in users, 58% [95% UI: 45–76%] in non-users and 37% [95% UI: 16–48%] at the community level when 80% of the population sleeps under a net. The larger indirect protection from the insecticide to non-users rather than to users is because users are already protected through the direct benefit of the barrier and the insecticide.

**Model predicted combined benefit.** The overall protection provided to one individual using an LLIN is estimated to be a relative reduction in EIR of 62% [95% UI: 36–74%] (sum of the above four different types of protection, Fig. 3e). This is over twice the reduction in personal EIR that the model predicts for one individual using an untreated net, which is 29% [95% UI: 12–53%] (Direct + indirect protection from barrier (Supplementary Fig. 4b)). The sensitivity analysis shows that unlike untreated nets there are no scenario when the use of LLINs increase malaria exposure to non-users (Fig. 3b). Protection provided to users and non-users of LLINs and untreated nets increases as their use increases in the community. When usage increases to 80%, the reduction in EIR for LLINs is 89% [95% UI: 67–98%] for users, 74% [95% UI: 48–92%] for non-users and 87% [95% UI: 63–96%] for the community (Fig. 3e). In contrast, the total protection offered by an untreated net used by 80% of the population is 42% [95% UI: 11–71%] reduction in EIR for users, 12% [95% UI: −3–40%] for non-users and 36% [95% UI: 8–65%] for the community (Supplementary Fig. 6b). In this scenario, the overall addition of the insecticide on LLINs accounts for 60% [95% UI: 32–87%] of the protection at the community-level. However, this relationship is non-linear (Supplementary Fig. 7), with lower usages providing a larger cumulative benefit than higher usages. It is interesting to note that the model predicts ~10% more protection is provided to people not sleeping under a net in a community with 80% LLIN use than would be provided by a single LLIN user in a community of non-users.

The specific results presented in Fig. 3 are for an initial pre-intervention EIR of 100 infectious bites per person per year. Absolute estimates and the relative difference between net types and users/non-users will vary with endemicity (baseline EIR) though similar trends are observed (Fig. 1b, Supplementary Fig. 8). The relative magnitude of the differences will also vary in low transmission settings as some intervention may cause local elimination. We see that the relative reduction

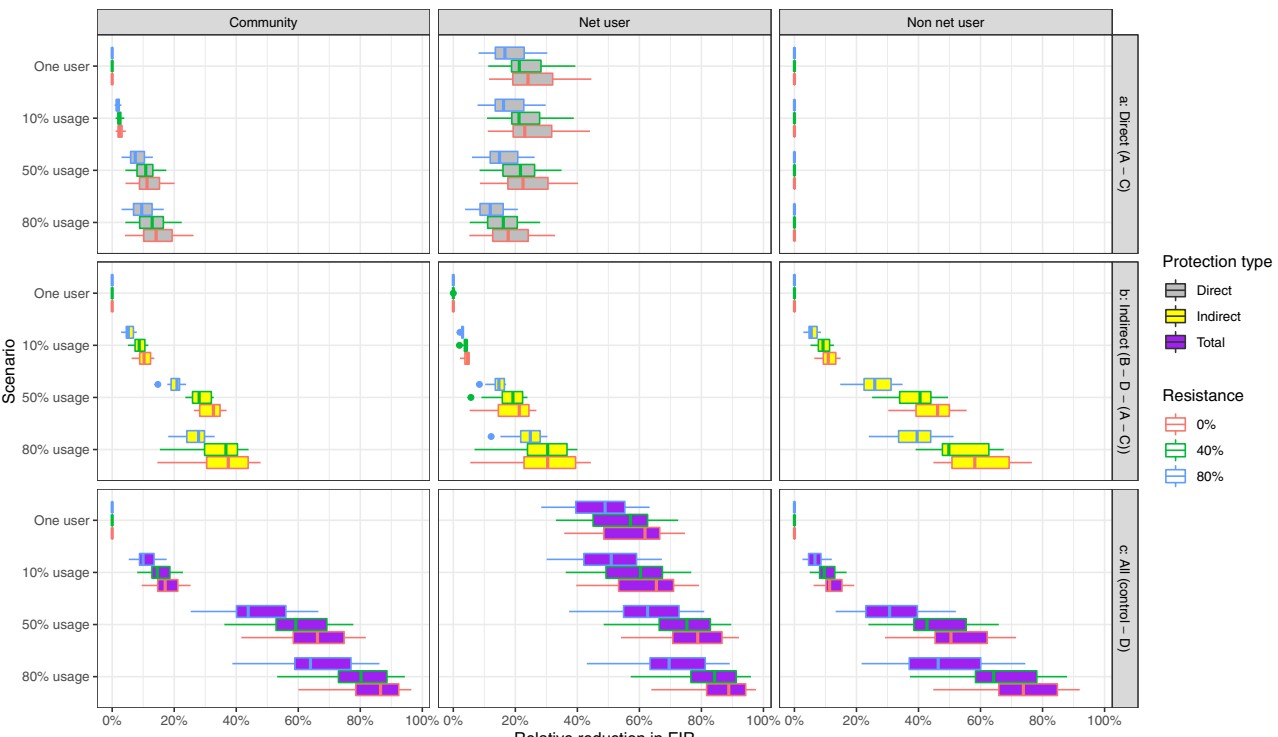

**Fig. 4 | Relative reduction in entomological inoculation rate (EIR) from mass community protection offered by LLIN at differing levels of pyrethroid resistance for a pre-intervention EIR of 100.** Left figure shows the effect for the community, middle figure shows the effect for users and right figure shows the effect for non-users. Rows (**a**–**c**) indicate type of protection offered. The reduction in EIR is calculated from a control, where nobody in the population is given a net, for the scenarios detailed in the methods section where insecticide is included. Usages of 1 person in the population, 10%, 50% and 80% are considered for 0% resistance (as in Fig. 3) and 40% and 80% resistance. The barrier only effects remain constant at differing levels of resistance so are not included. 15 samples were used to generate each box. The centre of the box and whisker plots shows the median, outside lines showing the first and third quartiles, and the whiskers indicate 1.5 times the interquartile range. Dots indicate outliers.

in EIR is on average slightly higher for an initial EIR of 10 than an initial EIR of 100. In addition to EIR, we are also able to tease apart the protection offered to users and non-users through the resulting reduction in all-age prevalence (Supplementary Fig. 9). The patterns seen for changes in EIR are similar to those found for prevalence.

## Types of protection from LLINs in the context of pyrethroid resistance

Lastly, we consider how these four components (direct benefit of barrier, indirect benefit of barrier, direct benefit from insecticide and indirect benefit from insecticide) vary with increasing pyrethroid resistance in the local mosquito population (Fig. 4). The direct and indirect benefits of the barrier do not change with resistance because the barrier mode of action is not impacted by pyrethroid resistance. We see that as pyrethroid resistance increases the additional direct impact of the insecticide for LLIN-users decreases as does the additional indirect benefit of the insecticide for LLIN-users and non-users.

For example, in the scenario shown in Fig. 4 we start with an EIR of 100 infectious bites per person per year with no pyrethroid resistance in the local mosquito population and 50% of people are using nets. The additional direct benefit of the insecticide for a LLIN-user is a relative reduction in EIR of 22% [95% UI: 9–39%] bites per person per year, but this falls to only an relative reduction of 15% [95% UI: 6–26%] bites per person per year when the local mosquito population exhibits 80% levels of pyrethroid resistance (as defined by survival of a discriminating dose bioassay, Fig. 4a). In addition, at the same usage level, the additional indirect benefit of insecticide is a relative reduction in EIR of 21% [95% UI: 6–27%] bites per person per year in the absence of pyrethroid resistant mosquitoes for net users, 46% [95% UI: 31–55%] for non-users and 33% [95% UI: 26–37%] for the whole community. This

falls to a relative reduction in EIR of 15% [95% UI: 9–17%], 26 [95% UI: 16–35] and 21% [95% UI: 15–24%] bites per person per year respectively at 80% pyrethroid resistance in local mosquitoes (Fig. 4b). Overall, at 50% net usage the insecticide causes 68% [95% UI: 46–89%] of the protection provided by LLINs, with this value reducing to 62% [95% UI: 42–82%] in areas with 80% resistance. Similar patterns are seen between users and non-users for different levels of LLIN usage.

Overall, the relationship between the level of resistance and protection is non-linear, with greater relative reductions in EIR seen at lower levels of resistance in the local mosquito population (Fig. 4, Supplementary Fig. 10). Again, the trends are similar for other initial EIRs (Supplementary Fig. 11) and for all-age malaria prevalence (Supplementary Fig. 12).

## Discussion

This work adds to the existing evidence-base for the presence of a community effect. Many experimental hut trial have shown that mosquito feeding attempts are altered by the presence of insecticide treated nets in rooms[34,42–44] and randomised control trials demonstrate that this altered behaviour—including increased mosquito mortality—is leading to significant public health benefits from nets[13,21,45]. Early trials indicated that non-users were also afforded protection from these actions on mosquito densities and behaviours[45,46]. Hawley et al.[13] saw a similar odds ratio for clinical malaria and other malaria indicators in control compounds without bednets <300 m from a border of compounds with nets. A review by Shaukat et al.[47] identified six studies that involved insecticide treated nets, but information about all-age usage levels were incomplete. For these studies percentage reductions in EIR were between −42% and 97% with all but one between 55 and 97%. Lindblade et al.[24] found there was a 91% reduction in EIR for a

65.9% usage in children under 5. It was our aim here to try to disentangle and quantify the direct and indirect protection offered from LLINs to both users and non-users into benefits afforded by the barrier distinctly to benefits from the insecticide. This allows us to estimate what is lost through the presence of pyrethroid resistant local mosquitoes that reduces the well-established critical role of the killing effect of insecticide-treated nets[12,15,17,18].

There is no direct method of exploring how reliable our mechanistic modelling results are so we use DHS data to explore how well it can reflect differences in the RDT prevalence of malaria between those individuals who reported they did or did not sleep beneath nets the previous night. Taking the lowest possible spatial scale (clusters), we can see that the broad qualitative conclusions identified by the mechanistic model are statistically supported by the observed data. However, we were unable to capture the heterogeneities within clusters that are inherent in natural field settings and likely to be expressed within the DHS data. Here we hypothesise this is driving some of the inconsistencies. Our model predictions show that LLIN users on average have a lower malaria parasite prevalence than non-LLIN users within the same clusters, but this difference is relatively modest, only becoming substantial when approximately half the people in the cluster are RDT positive. Similar trends were seen in the observed data, with only a significant difference between users and non-users in clusters with intermediate disease endemicity. This result is also consistent with recent studies that saw similar patterns between users and non-users[48–50]. Both the mechanistic model and the observed data suggested that the level of net use does not influence the difference between users and non-users. Similarly, our model predictions show very little difference between users and non-users in the presence of insecticide resistance and we could not see a change in prevalence of the two groups over time or with increasing pyrethroid resistance in local mosquitoes in the empirical data. The similarities of the model and the observed data afford some comfort that the model assumptions are appropriate. Nevertheless, care should be taken interpreting results as we are not able to directly measure the different benefits of LLINs, and the deterministic model we employ tends to overestimate the difference in prevalence in LLIN users and non-LLIN users relative to the DHS data. This difference is likely due to the variability in real world data that can stem from multiple sources, and we do not calibrate the model simulations to each cluster individually. For example, the local ecologies of each cluster will likely alter the potential for transmission given ratios of mosquitoes to people associated with rainfall, temperature, and land-use[51–53], the preference of local mosquitoes to bite during the night, on people or indoors[54–57]. The use of nets over seasons may well cycle with peaks in net use occurring when the risk of transmission is greatest[58], and net use may also vary among community members of different ages[59]. Coupled with these socio-ecological reasons, the addition and use of complementary interventions has altered over time which is likely to contribute further to variability in these DHS data.

A limitation of our model is that we assume systematic use of LLINs over time, i.e., users will always use a LLIN each night whilst non-users will not, though community use is simulated to wane with time since the previous mass campaign. In reality, use will vary from night to night, which reduces our ability to detect a difference in malaria burden between users and non-users. This variable usage does not theoretically influence the direct and indirect benefit of nets, just our ability to detect it in the survey data. There is currently no method of assessing this variability from the DHS data for each cluster which is designed as a cross sectional survey asking whether a child slept under a net the previous night (although other unmatched data exists e.g. ref. [60]). In addition, in our model we assume LLINs are distributed at random within the community, whereas people may be more likely to use nets in regions of the cluster more prone to mosquito biting (for example, nearer breeding sites), further potentially minimising the difference between users and non-users. Seasonality of transmission is also ignored given the complexity to untangle the direct and indirect types of protection afforded by nets. On this basis, we assume a constant level of exposure to mosquitoes in the presence of mosquito nets throughout the year. Evidence suggests that net use fluctuates annually[60], and that exposure to infectious bites may be equally variable given human activity within a community and mosquito activity seasonally[61]. Parameterising an individual-based model with field estimates of this inter and intra season, nightly, and within-cluster variability could substantially reduce the discrepancy between observed data and model predictions and could enable us to capture within cluster heterogeneities. This could be possible through re-analysis of results from early cluster randomised control trials where heterogeneity could be quantified before and after the introduction of ITNs. Since our model assumes systematic net usage, the estimated difference between ITN users and non-users is likely a maximum, with heterogenous usage and transmission likely to diminish this difference in practice. So, though the model can capture the broad qualitative conclusions, the absolute magnitude of these differences should be treated with caution. Nevertheless, whether the model captures these differences should also not substantially influence our quantification of the direct and indirect benefits of LLINs, which rely on similar structural assumptions, just our ability to match the model results to field data.

The modelling framework allows us to untangle the community protection offered from the barrier and insecticide, which stresses the additional benefit elicited from killing mosquitoes and supports other findings in literature[12,13,31]. Importantly we parameterise the model with results from experimental hut trials evaluating both insecticide treated and untreated nets as the physical barrier of the mosquito net has been shown to induce some mortality in mosquitoes as they attempt to feed[34]. Nevertheless, we estimate the relative protection provided by the insecticide is considerable, the relative reduction in EIR for LLINs is 89% [95% UI: 67–98%] for users and 74% [95% UI: 48–92%] for non-users in areas in our hypothetical setting (simulating an EIR of 100 prior to the introduction of 80% net use). This is important because in places with mosquitoes that are resistant to the pyrethroid active ingredient on the nets we might expect an intermediate effect size to these two extremes. Our models quantify resistance according to the percentage of mosquitoes surviving 24 h following exposure to a discriminating dose bioassay. This assay is relatively unsensitive and further work is needed with more reliable phenotypic or genetic assays, potentially considering sublethal effects of the insecticide on the mosquito. The work indicates the value of focusing on insecticidal potency[12], or the potential need to couple mosquito nets with a second intervention that kills local mosquitoes (like IRS[62,63] or that focuses on reducing their numbers like larval source management[64,65]).

Results here and elsewhere[9,12,29,66] illustrate the importance of achieving high coverage with both LLINs and untreated nets as increasing net use generally will be beneficial for community protection (Supplementary Fig. 7). Killeen[12] argues that prioritising vector mortality rather than universal usage may be more cost-effective if newer products, with alternative active ingredients, are more expensive (and therefore fewer can be procured for a given region). This makes sense if the killing effect is particularly potent, and nets are distributed in a way that means non-users are in close enough proximity to benefit from the reduced numbers of mosquitoes caused by the killing action of the net. Novel net designs that either chemically[67–69] or mechanically[70] kill mosquitoes are welcomed in order to mitigate for the potential loss in impact from pyrethroid-LLINs[33,35]. It may be cost-effective to then top-up communities with untreated nets to provide some barrier protection to those members who do not have the treated net. A cost-effective analysis is beyond the scope of the current work, particularly because it becomes logistically harder and more expensive to increase usage once net use has reached

around 60% so the association is non-linear[4], but certainly worth consideration for future discussion.

There was no measurable difference between disease prevalence in users and non-users in the DHS data over time or according to the level of pyrethroid resistance in the local mosquito population. Previous studies have suggested that higher infection rates in non-users than users seen in recent data may indicate reduced community protection[48]. Our analysis of DHS data does not support this suggestion as there is no evidence that the difference between users has changed over the time-period during which resistance has spread. The analytical work provides a reason for why this might be the case as though the model predicts a reduction in the difference between users and non-users as pyrethroid resistance increases (Supplementary Fig. 5) the magnitude of this change is relatively modest in relation to the differences caused by the barrier effect of the net. Care should be taken interpreting the DHS data combined with our estimates of the level of resistance in a community using the discriminating dose bioassay because the resistance estimates are highly variable and have been shown to be heterogeneous across the districts. The level of resistance in each DHS cluster was not directly measured but instead inferred, reducing the strength of the analyses. Thomas and Read[71] suggest that it is the indirect, or community, protection that is at greatest risk with increasing pyrethroid resistance for countries with moderate-to-high prevalence but incomplete ITN coverage. This statement is supported by our theoretical findings. Our modelling results further suggest that it is both the direct and indirect impact of the insecticide that are impacted by pyrethroid resistance, but the impact on the indirect effect is larger (Fig. 4, Supplementary Fig. 10, Supplementary Fig. 11 and Supplementary Fig. 12).

It should be stressed that these modelling results are dependent on the assumptions made in the modelling process (for example, what happens to a mosquito when they are dissuaded from feeding on someone who is using a net, outlined in full in the supplement). Though these assumptions reflect our current best understanding of the processes involved, they should be verified, for example using recently developed technology that can track the mosquito in the home[72]. In our simulations, the direct and indirect benefits of the barrier do not change with resistance because the barrier mode of action is not impacted by pyrethroid resistance. This assumes that net brands have equivalent integrity whether they are untreated or treated with pyrethroid insecticide. Recent work indicates this may not be the case[36] which needs further investigation as net durability is a critical part of community protection[12]. Maximum direct protection from indoor interventions is limited in places with higher levels of outdoor biting (be it biting during the early evening, later morning or daytime) even if no pyrethroid resistance exists and everyone uses LLINs[56]. The indirect protection provided by the insecticide can help overcome the issues of residual transmission though this benefit is diminished as mosquitoes develop resistance to the insecticide[56,73]. Determining the different protective effects of LLINs allows us to better understand how insecticide resistance is likely to impact on malaria control, and how best to mitigate against it. Appreciating all aspects of protection that are afforded by LLINs will be crucial in the drive for malaria control —helping to shape how we can innovate nets with improved effectiveness—particularly given the likely reality of insecticide resistance.

## Methods

### Demographic health survey data comparison
We use Demographic and Health survey (DHS) data to evaluate the difference in prevalence in 6–59-month-olds for clusters with different prevalence and usage levels. We include all available DHS data for African countries since 2010 with geolocated cluster data, *Plasmodium falciparum* malaria prevalence and LLIN usage[41]. In these DHS data, a cluster typically represent a census enumeration area, which can be villages in rural areas or city blocks in urban areas and represents as

close to a homogenous area as possible. The total number of people of all ages in each cluster varies from 40 to 543. Net usage is estimated from the household survey by asking a household member of age 15 years or older, who in the household slept beneath a LLIN the previous night. DHS surveys have evolved over time and this question has been addressed to different household members. A total of 47 surveys from 21 countries are included; they are listed in Supplementary Table 2. We use the *rdhs* package to extract data[74]. We only consider usage and RDT outcomes from clusters with 20 or more children, which results in the inclusion of 111,250 individuals of 6–59-months of age in 4138 clusters.

We developed a linear mixed effects model from the R package lme4[75] to investigate what was driving the difference between prevalence in non-users and users (outcome variable). Following visual inspection of the mechanistic model outputs, clusters were grouped into low (0–33%), medium (33–66%) and high (66–100%) malaria prevalence to allow overall trends to be identified. In the main text net usage within the cluster was restricted to low (20–50%) and high (50–80%) net usage to reflect the ranges normally seen in country programmes. Alongside cluster usage and prevalence group we included an interaction between the prevalence and usage grouping and a country specific random effect. Uncertainty was calculated using the bootMer package where 1000 bootstrap samples were taken for each data point.

Pyrethroid resistance, as approximated using susceptibility bioassays, has increased across many areas of sub-Saharan Africa over time[76]. Evidence is building that suggests the presence of mosquitoes with resistance to pyrethroid insecticides may reduce the efficacy of mosquito nets[32–35,77]. Therefore, we wanted to explore whether we could see an effect of pyrethroid resistance on the difference between malaria parasite prevalence in net users and non-users. We used a systematic review of susceptibility bioassays collated from 2000–2018 across the African continent[76,78] to investigate changes in resistance. For this linear model, we used a simplified model with the difference in prevalence between non-users and users as the outcome variable aggregated to the first administrative level (often the District or Province level of a country). However, we used pyrethroid resistance (approximated as the proportion of mosquitoes surviving exposure at susceptibility bioassay testing) instead as the predictor variable. Our data consisted of 357 observations across 16 countries. We were able to estimate this predictor of resistance at the administrative 1 level by fitting logistic functions to the available data from each country.

Clearly, there are limitations with this broad-brush approach, not least given the variability in both the assay and geographic expressions of resistance[34,79,80] as well as varying prevalence over time and space— we present these limitations in the discussion—but we include the analysis to see whether there is a signal in the empirical data, and to give some context for our subsequent modelling exploration of what increasing resistance may mean for continued protection offered by a community effect.

### Modelling approach
The difference in user and non-user prevalence from the DHS data are compared to model predictions from a deterministic version of a well-established compartmental *Plasmodium falciparum* malaria transmission model[37]. Depending on the analyses both all-age prevalence and 6–59-month old prevalence are presented. This model incorporates a full dynamic mosquito-vector component[38], which satisfactorily represents vector-control interventions[81]. We briefly describe the model below focusing on how the effect of LLINs are incorporated, but the full mathematical detail is provided in the Supplementary Methods and the following references[37,39,40]: For this study we assume a well-mixed population with no seasonality.

The community considered is split into age, heterogeneity, and intervention compartments. Each age compartment is assigned a

unique biting rate with the parameters that determine the relationship between age (i.e., body size) and biting rate described in the Supplementary Methods. In this model we consider two intervention groups, those who sleep under an LLIN (LLIN users) and those who do not (non-LLIN users). We assume a LLIN user always uses their net, with a 3-year replacement cycle and that in the absence of control interventions 85% of mosquito bites are taken on people when they are in bed (estimates with LLINs will be higher and depend on net type, usage and time since deployment). The population is initially modelled as susceptible, with people moving to the infected state at a given rate through bites from infectious mosquitoes. Following a period reflecting liver-stage infection, a proportion of the individuals in the infected compartment become symptomatic and seek treatment. The proportion who receives successful treatment experience a period of drug-dependent prophylaxis before returning to the susceptible compartment. The proportion of symptomatic individuals who do not receive treatment experience a period of symptomatic disease before transitioning into the asymptomatic compartment. The proportion of the population who are asymptomatic move from patent infection to sub-patent before natural recovery transitions them back to the susceptible compartment. We also include superinfection in the model from asymptomatic and sub-patent states. Superinfection is defined here as an infected person who can be bitten and infected again by an infectious mosquito.

We consider three types of immunity in our model, with parameters estimated by fitting to clinical incidence and parasite prevalence data[37,82]. Clinical immunity is exposure-driven, so age dependent, and protects the population against clinical disease. Anti-parasitic immunity develops through age and exposure to infection and reduces how easily infections are detected through the control of parasite density. Anti-infection immunity develops later in life and reduces the probability that an infectious bite results in patent infection. All human infection states are assumed to be infectious to mosquitoes, with infectivity correlated with parasite density.

The compartmental vector model has larval stages and adult female mosquitoes similar to Griffin et al.[37] and White et al.[38] This results in the following entomological inoculation rate (EIR),

$$EIR_k = I_M \lambda_k \tag{1}$$

where $I_M$ is the number of infectious mosquitoes and $\lambda_k$ is the intervention compartment-dependent rate at which a person is bitten by a mosquito, which varies depending on the LLIN usage and pyrethroid resistance values ($k = 1$ for non-users and 2 for users). The probability of a blood-seeking mosquito successfully feeding depends on the behaviour of the mosquito and the anti-vectoral defences employed by the human host population. There are three possible outcomes tracked in the model once a mosquito enters a house to feed: it can repeat ($r_N$), feed successfully ($s_N$) or die ($d_N$). Over time, the efficacy of the repellence effects of the LLIN fluctuates from a maximum, $r_{N0}$, to a non-zero level, $r_{NM}$, which reflects the protection still provided by a net that no longer has any insecticidal effect (and potentially some holes). The killing effect of ITNs decrease from a maximum $d_{N0}$ when the insecticide is working optimally. The outcomes $r_N$, $d_N$ and $s_N$ always sum to 1. These estimates will be altered in the presence of mosquitoes that are resistant to pyrethroid (see Supplementary Methods).

The net model has been parameterised using data from a systematic review of experimental hut studies[35]. This review collated 90 experimental hut trial arms with an untreated net included as the control. These data are used to parameterise the benefit of an untreated net relative to a no-net control adopting the same method as LLINs. This is important because untreated nets are also thought to increase mosquito mortality relative to a no-net control, so prescribing all mortality due to the action of the insecticide is likely to overestimate insecticidal impact. Briefly, the entomological impact of

LLINs is tested in huts to estimate the probability that a mosquito will successfully blood feed, be killed, or continue on without feeding (a combination of deterrence; not entering the LLIN hut at all, and repellence; entering and then exiting without feeding). In the absence of a mosquito net, it is assumed that 69.9% of mosquitoes will successfully feed[66,83] and otherwise exit. The efficacy of the killing effect measured from LLIN experimental hut trials is associated with mosquito mortality in the discriminatory dose susceptibility bioassay tests that are used to approximate any changes in pyrethroid resistance over time for a given location. The associations between experimental hut mortality, successful blood-feeding and deterrence, that are driven by LLIN presence, are also predictably altered as the proportion of mosquitoes surviving a susceptibility bioassay test increases[35]. Therefore, these associations can be used to estimate the lost impact of LLINs in the presence of pyrethroid resistant mosquito populations. Net use is assumed to be random within the population and we assume systematic use of nets over time (LLIN-user sleep under LLINs every night).

When comparing the model data to the DHS data, we used bisection to fit the initial EIR (model input number) to match the post intervention prevalence of 0.2–0.8 in children aged 6–59-months-old for LLIN usage levels of 20–80%. We used a simple logistic regression to show the closeness of the estimates from the mechanistic model (outcome variable) with data (predictor variable).

### Direct vs indirect (mass community) protection

Since it is hard to address the different types of protections offered to users and non-users from data, we used our transmission model to tease apart the direct and indirect (mass community effect) protection offered by LLINs. In addition to a control where nobody in the community is given any nets, we identify 4 scenarios that describe the different types of protection offered by untreated nets or LLINs to users and non-users (Fig. 2). We fix the EIR over time in some scenarios (by breaking the link between current malaria endemicity and the human force of infection) to disentangle the direct and mass community effect of nets. This is implemented in the model by fixing the number of infectious mosquitoes (Eq. (1)) to the number in the control and fixing the biting rate of non-users ($k = 1$) to the biting rate in the control scenario.

A. Direct protection from barrier to x% of the population. x% of people protected with an untreated net for a fixed EIR.
B. Direct + indirect protection from barrier to x% of the population. x% of people protected with an untreated net for a varying EIR.
C. Direct protection from barrier and insecticide to x% of the population. x% of people protected with a LLIN for a fixed EIR.
D. Direct + indirect protection from barrier and insecticide to x% of the population. x% of people protected with a LLIN for a varying EIR.

We illustrate the impact of the mass community effect by investigating the reduction in both EIR and all-age prevalence from the control scenario for users and non-users for hypothetical non-seasonal settings with pre-intervention EIRs of 100 and 10. We repeat scenarios (A)–(D) for four different usages: an individual (1/100,000), 10%, 50% and 80% (the maximum usage expected in a community—that is, the proportion of people using a mosquito net at deployment immediately after the mass net distribution) to investigate how the mass community effect varies with usage. In each scenario, we run the model for 1 year and then add the intervention. Initially, we subtract the average of the EIR or prevalence between years 3 and 6 from the control (to average over the three-year waning cycle; nets given in year 1, 4, 7), depending on the observation so we are presenting a reduction in EIR or prevalence due to the intervention. We then calculate the reduction in the indirect part of the protection offered by subtracting the direct protection from the direct + indirect protection, for example the

 

indirect protection from the barrier for an individual is scenario $control - B - (control - A)$ or $A - B$. Further, we calculate the additional benefit of the insecticide by subtracting the protection from a scenario without insecticide from the similar scenario with insecticide, for example the additional direct protection from insecticide to an individual is $control - C - (control - A)$ or $A - C$, see Table 1.

We present uncertainty in the estimates from our mechanistic model by varying the following important parameters: the proportion of mosquitoes that repeat and the proportion of mosquitoes that die for LLINs and untreated nets and the proportion of bites taken on humans in bed (Supplementary Table 1, Supplementary Methods). Other parameters that influence both users and non-users, such as the time spent by a mosquito looking for a blood-meal, are kept constant. There is a wide range of parameter uncertainty for the efficacy of the different nets and how LLINs are influenced by pyrethroid resistance. Here we assume that the efficacy of LLINs can be no worse than untreated nets. This assumption should be verified with naturally aged nets from the field, as different net brands might physically deteriorate at different rates[36], though in the absence of data on brands of nets used in various locations, this assumption seems the most parsimonious. Uncertainty in our LLIN parameterisations is carried through from the analysis of empirical data studies using experimental huts[34]. One thousand posterior parameter draws are taken from statistical fits to the empirical data measuring mortality, successful feeding, and deterrence[34]. The data are collated following Griffin[37] to provide ranges for uncertainty in net efficacy parameters (Supplementary Table 1). The median and 90% credible intervals of this range are then included for the sensitivity analysis. We investigated how the direct and mass community effect of LLINs might change with increasing levels of pyrethroid resistance in the local mosquito population.

### Reporting summary

Further information on research design is available in the Nature Portfolio Reporting Summary linked to this article.

## Data availability

The Demographic Health Survey data is publicly available at https://dhsprogram.com.

## Code availability

The deterministic malaria model used for this analysis can be found on GitHub (https://github.com/mrc-ide/deterministic-malaria-model)[84] along with the scripts for our analysis at (https://github.com/ettieunwin/direct_indirect_bednet_protection)[85].

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

## Acknowledgements

H.J.T.U. and A.C.G. were funded by a grant from the Bill and Melinda Gates foundation, and HTU additionally from an Imperial College Research Fellowship. E.S.-S. is funded by a UKRI Future Leaders Fellowship from the Medical Research Council (MR/T041986/1). Funding support for T.S.C. and E.S.-S. was also received from the Innovative Vector Control Consortium through the New Nets Project funded by Unitaid, the Wellcome Trust [200222/Z/15/Z] MiRA. All authors acknowledge funding from the MRC Centre for Global Infectious Disease Analysis (reference MR/R015600/1), jointly funded by the UK Medical Research Council (MRC) and the UK Foreign, Commonwealth & Development Office (FCDO), under the MRC/FCDO Concordat agreement and is also part of the EDCTP2 programme supported by the European Union.

## Author contributions

H.J.T.U. and E.S.-S. designed the study, conducted the analysis, and wrote the first draft of the manuscript with guidance from T.S.C. and A.C.G. All authors commented on the final manuscript.

## Competing interests

The authors declare no competing interests.
