## [Peer Review File · Nature Communications]

Quantifying the direct and indirect protection provided by insecticide treated bed nets against malariaREVIEWER COMMENTS

Reviewer #1 (Remarks to the Author):

The manuscript attempts to unite modelling / theoretical work with data analysis from DHS datasets to describe and quantify and compartmentalize the various elements of the 'community' effect of ITNs/LLINs for malaria. This is an ambitious work, partly because while the idea of a community effect is widely discussed, it is understood (or mis-understood), measured and discussed in varied and inconsistent ways throughout the malaria literature.

The work appears to generally be sensible and the approach to doing reasonable especially on the modeling side. Though the analysis and the role of the DHS/observational data in supporting this work seems less clear, and not well connected to the large amounts literature on the effect of ITNs using real world data.

Overall i think the manuscript is important and original and highlights some very important areas in which the community effect of ITNs is misunderstood/poorly quantified and could be helpful for malaria epidemiologists, and modelers health economists and in strategic planning for malaria control programs.

Major concerns:

1) The methods for the analysis of DHS data are not clearly reported, and some descriptions of these data are not correct:

1a) in DHS data a 'cluster' is typically an census enumeration area, which may be as small as a city (or a few) block or part of a village, and has nothing to do with what health facilities are used for treatment (line 355)

1b) net usage is not ascertained from head of household. In the current versions of DHS questionnaires this IS part of the household questionnaire, but this questionnaire can be answered b anyone of the age 15 or older in the household. The ways and from whom these questions were asked have changed over time.

1c) the question is not asked by person it is part of the net roster in which it is asked who on the household roster slept under a specific net the night before the survey.

1d) on line 355 a data aggregation to admin 1 is described. Here admin 1 is described as district or province, but admin 1 is generally province or region, district is typically and admin 2 disaggregation.

1c) DHS data follows a complex multistage sampling and stratification and aggregation of data to levels for which the sample was not designed to represent can lead to misleading conclusions. It is not clear why the authors did not use DHS stratifications in this component of the analysis.

1d) line 362, what trends are the authors referring in the linear regression? net use? prevalence? resistance predicted by the other? it is not clear from the methods what exactly this analysis is?

1e) given the restriction of having at least 5 children using and 5 not using nets in each cluster of inclusion, surely the distribution of clusters is biased away from extreme use/not use clusters in this analysis as well as from small clusters.

2) The comparison between modelled data and DHS data seems to be completely qualitative. This is probably/possibly all that is doable in this particular framework, but the methods don't discuss any approach for this comparison or how the conclusion that the data supports the authors hypothesis.

2a) line 97 - "the hypothesis that any difference in malaria burden in users and non-users is not influenced by usage" Is a very unclear sentence. As written it seems tautological, since the

difference between groups created by conditioning on usage, must be due to usage? I think the authors mean that the absolute difference in prevalence between the groups of users vs. non-users is not influenced by community levels of usage? If so there are straightforward statistical methods for making such assessments such as the testing of interaction terms. I'm a bit surprised that the authors have not chosen to undertake such an approach, or really applied statistical methods to this data at all.

2b) line 104 - similarly the authors state that they did not "detect a signal" that the difference in prevalence between users and non-users changed over time, or with level of resistance. but don't clearly describe the method used to search for this signal. I believe that this is a linear regression aggregated to the admin one level, for resistance, though for change over time it is less clear. Regardless neither the outcomes nor the exposure variables, the dataset size etc... are fully described in the methods for this.

3) the authors focus very much on presenting results as reductions in EIR to users vs. non-users and throughout don't present overall reductions in EIR consistently. While the disaggregated approach is helpful for highlighting the difference, it also sort of masks the overall meaning of this effect on the outcomes which are of the most clinical and public health meaning, which are reductions in EIR at the community level or reductions in parasite burden at the community level. because the importance of the protection of users vs. non-users changes and the prevalence of use changes, its important to think about community protection too.

4) a sort of follow on to number 3 is that the authors suggest at least twice in the manuscript that the implications of the analysis are that more protection is always favored. This is only true when one considers 'more protection' as unconstrained maximization of health benefits with no cost function or alternative. This is a view that is, at least I think, misleadingly simple. The authors have done the hard work of essentially estimating the effects of varying coverage, including the mass effects of nets, but completely neglect to describe or consider the marginal gains made by increasing coverage which can be estimated from the outputs here (or to even mention that such gains inevitably come at a cost which will be a tradeoff against something).

4a) I wondered also why in the interest of symmetry the authors didn't model an even higher coverage scenario, such as nearing 100% usage of nets. While these are unlikely to be achieved in real life, this might help fill the whole spectrum of net usage. Despite the analysis restrictions imposed by the inclusion criteria the authors have chosen to model a scenario at the low usage end which cannot be found in real world data. One wonders why not one at the high end.

5) On line 279-280: the authors make a statement "there is no minimum LLIN usage needed for users and non-users to benefit from community protection" This statement, which i think the paper supports, and I also think is true is a point which is missed by a huge proportion of the community and might deserve more emphasis. The idea that there is some threshold coverage that must be achieved for community protection is widely held, and the idea that community protection benefits not only non-users but also other users is also often seems to be missed. the authors might make this a clear point of their findings

More minor comments:

1) throughout the main text and supplementary information the tables and figures sometimes lack complete notation descriptions making them difficult to interpret as stand alone documents: examples include in the supplementary information, the human compartment model diagram, P is not described nor are any of the flow parameters, in the first table illustrating the comparisons the a , b,c, d, and control notation is used, but without a caption/note to help the reader understand these. Which requires cross reference to text which may not be easy to find in final production. Etc...

2) line 224- "the difference is because of the innate variability in real world data" doesn't mean much.

line 228 - what is meant by 'sampling bias' and how is the exacerbating variation? Bias generally means bias, and is independent of sample size. Variance or variation, while also independent of sample size directly integrates into precision which is affected by sample size? also this is an incomplete sentence.

line 271-273 : increasing net use always increases protection... again ignores the question of whether or not this increase is meaningful. It a pretty limited statement to note that the relationship between net use and impact is monotonic, describing the shape of this function (and its first derivative) would be of use especially in priority setting.

a minor technical point I think that Insecticide resistance is parameterized throughout as simply decreased killing of mosquitoes, but in practice there may also be changes to the repellency/deterrence. It may be worth noting that this may not fully capture the nuances of IR.

line 290 and 291: its a bit confusing to say maximize the use of LLIN then increase coverage with untreated nets. I think in practice it may actually be relevant, since there are populations that may never get/use government issued LLIN but could procure untreated nets on the market (such as in urban areas) but this reads as if you are advocating for 100% usage of LLIN then adding untreated nets.

line 306-307: inferring the level of insecticide resistance reduces the 'power' of the analysis? How? This seems an odd conclusion. Also how do you know that the dataset is underpowered to detect the clinical differences suggest by the model, have you done a formal power analysis to justify this?

Reviewer #2 (Remarks to the Author):

The authors explore an interesting question of attributing the individual and community effects of bednets, further distinguishing the effects of the bednet barrier vs insecticide, and consider the impact of both bednet coverage and insecticide resistance on modulating these effects. They use a deterministic modeling approach with parameterization based on statistical models fitted to experimental hut trial data. The authors also compare prevalence in net users and non-users at the cluster level using DHS surveys across multiple countries, years, and epidemiological contexts.

Gathering quantitative evidence of the individual- and community-level effect of ITNs is important and helpful for decision-makers, especially in the context of insecticide resistance and continued need for effective vector control. However, I felt that the work as presented was a missed opportunity to use a modeling platform to more deeply explore questions around ITN effect size that are extremely challenging to measure in the field. Furthermore, several aspects of the presentation were confusing and would benefit from clarification.

Major comments:

1. It is unclear what to make of seeing both the DHS and modeling results together. They don't seem to be in agreement beyond both seeing the same direction of effect (a low bar). The authors list reasons why this could be, but choose not to investigate any of them (perhaps because of major comment #2 below). At the end of the day, I'm not sure why I should believe any of the model results are relevant to the real world since they are not reconciled to actual measurements. It would still be interesting to look at this question of ITN effect size attribution in an abstracted model, but the authors' framing is very much toward quantitatively assigning these attributions rather than a general exploration of the relative magnitude of effects.
2. It seems that choosing to use a deterministic compartmental model rather than an individual-based model has really limited the authors' ability to investigate their research question in more detail. For example, imperfect compliance to net usage (individuals occasionally do not use their net), heterogeneity in exposure (correlated, uncorrelated, or anti-correlated with net usage), and

spatial effects could each contribute to the attenuated difference observed in DHS vs in the model. The authors could potentially evaluate the possible contribution of these types of secondary (or primary?) effects in a modeling framework, and this work would be both more interesting and more relevant.

Minor comments:

1. Please add page numbers, this will make it easier for reviewers as we flip back and forth.
2. L67-68: this is confusing. What is "this protection" that L67 is referring to? The barrier? The net in general? The insecticide? Are you then saying that to the individual-level protection there is added i) physical barrier protection, which I would think would already be included in individual-level protection?
3. L97: "the trends match" could you elaborate on this and explain exactly what is meant?
4. L111: what is meant by "in the absence of data"? what kind of data?
5. L121: clarify that in this artificial scenario, you enforce that EIR on the non-users is unchanged. I found the framing of the "direct benefit" scenarios (EIR on users is allowed to respond to the presence of the net, but EIR on non-users is forced to maintain original value) unclear on first read-through of this section. I figured it out after digging through the methods, but it would be helpful to have an explanation of this approach to EIR also stated in the results. I found the term "fixed EIR" confusing since EIR is only fixed on the non-user population, and "varying EIR" confusing since I would interpret that as "seasonally-varying".
6. L129-133 and elsewhere: I found the concept of "there are more bites because a vector deterred by a barrier will attempt to bite again, and the more net-users there are, the more this happens" to be confusing. Are the total number of bites not the same if there are nets vs no nets (each vector feeds at most once)? Is this a real phenomenon or an artefact of the model? What happens if you change the model so that this doesn't happen? It feels like there's a lot of airtime given to this concept, possibly more than it deserves given that we don't know if it's real (?) AND that the direct protection from the barrier is stronger than the increase in biting.
7. L138: "more mosquitoes are killed" is this a meaningful increase for the barrier-only bednet?
8. L174: "to a single person" unclear if this means per individual or if it means in the situation where only 1 person is using a net
9. L192: any key takeaways from the analysis on reduction in all-age prevalence?
10. L197: I thought repellency was part of the barrier model, does pyrethroid resistance not impact repellency?
11. L201 and others: what is a reduction in EIR of 22? Fig 4 caption says initial EIR = 10, how could it reduce by 22? Or is this 22%?
12. L207: "greater reductions" reductions in what?
13. L213-4 and L338-9: "the role that LLINs play in reducing mosquito populations cannot be underestimated" what does this mean? If the role cannot be underestimated, does that mean that it's very low to begin with? Or did you mean overestimated? Of course the role can be overestimated, we still have malaria. Not sure what's going on here.
14. L233: DHS will not report on variability in night-by-night usage but surely there are research studies that have looked at this. In fact some studies are cited a few sentences later, and general assumptions could be made from other data sources.
15. L265-9: is this consistent with empirical data?
16. L288-9: yes the insecticide is powerful but this sentence implies to me that the barrier has no effect. Perhaps reword
17. L307: "under-powered" is this meant in the technical sense of power? It seems not, in which case a different term would be more appropriate
18. L359: why RDT prevalence and not microscopy?
19. Methods: please also describe assumptions on seasonality (no seasonality), mixedness (well-mixed (?), and age effects on biting risk and ITN usage. LLINs are distributed at the beginning of years 1 and 4, is that right (Fig 1A suggests year 4 redistribution)? The year 4 event is not mentioned (L463).
20. L408-17: what are the units on lambda? Is I_M per person, or total over the population? How is lambda related to the r , d , and s parameters? Does a repeated bite attempt mean the mosquito ages a day between attempts or not?
21. L464: using "EIR/prevalence" to mean "EIR or prevalence, depending on which outcome we're observing" is confusing, it looks like you are dividing EIR and prevalence!

22. L670 Table 1: should be figure 2 instead of figure 1
23. L680-3 Fig 1B caption: is the x-axis EIR in the absence of nets? Or EIR in the subpopulation?
24. L707 Fig 3, S3, 4: would be easier to read values off these figures if the gridlines were at 10% intervals
25. L710: 5 scenarios, not 8, I believe
26. L707 Fig 3: I recommend splitting this into 5 panels A-E. In the results section you could then refer to specifically which panel you are discussing, which I for one would find quite helpful for navigating this figure.
27. L734 Table S1: the uncertainty ranges on the treated net parameters d_{NO} and r_{NO} seem fairly narrow, especially given that Nash et al. plots show really large variability in feeding success. Also: is net retention not modeled at all? If so, should note in methods.
28. L789 Fig S3: a lot of this is the same as Fig 3, right? But the yellow means something different than it did in Fig 3. Perhaps recolor the S3 yellow to something else, I was pretty confused what was different vs not between the 2 figures.
29. L809 Fig S4: it's not easy to look across these plots and spot differences. Why not pick a few most relevant of the scenario-coverage combinations and plot outputs vs EIR? Would be easier to assess the impact of EIR on effect size.

Reviewer #3 (Remarks to the Author):

MAJOR COMMENTS

1. In the abstract, the following statement is not entirely true and the quotes should be removed from the community effect, because it's a very real, well established phenomenon: "The inherent linkage between these types of protection renders it impossible to empirically quantify the 'community effect'." For example, see figure 2 in Hawley et al 2003 AJTMH 68(Suppl4): 121 for an example of more-or-less full community effects experienced by people lacking nets but living right beside the boundary with areas enjoying full coverage.
2. The bottom-line conclusions in the abstract represent nothing new and have been both modelled and demonstrated empirically many times over the years. Indeed, relevant principles (Killeen et al 2007 PLoS Med 4: e229) and supporting evidence (Pryce et al. Cochrane Database of Systematic Reviews 2018, CD000363) were accepted and adopted as WHO policy back in 2007 (<http://medi-guide.meditool.cn/ympdf/FBACA171-AF95-DCFA-673E-3FA3AFEB79AE.pdf>). While the modelling here looks solid and there's nothing incorrect about the conclusions, presenting them as new insights, rather than old but often ignored ones (Killeen 2020 Lancet 395: 1394), is misleading and I'm sure the authors know as much.
3. Similarly, statement like the following on line 52 of the introduction are simply inaccurate and seem to ignore all the fine details of all the many papers published over the years: "Evidence of the community effect is ambiguous, partly because empirical studies have not been powered to address this question – it would be unethical now to deliberately withhold nets from a cohort of a community." Indeed, it certainly would, but there are also many folks who simply decline to use them even when provided freely and many other examples where this information can be gleaned from field studies, most notably the spatially explicit analysis in Hawley et al 2003. Similarly, all the statements in the paragraph beginning on line 63 about community level effects being impossible to define, much less measure, are inaccurate and appear intended to over-represent the novelty of the work presented. Similarly, while the new analysis described in the final paragraph of the introduction is certainly valuable, the statement "which is not possible to do from trial data" goes too far.
4. The actual analysis presented in the results section is quite neat and impressive and does add some nice depth and nuance to the evidence base, so this remains a useful paper worthy of publication, but the abstract and introduction need to be reframed around what this study does add rather than exaggerate the novelty of simply distinguishing community and personal protection effects.
5. Lines 56 onwards: The apparent lack of an obvious community effect in The Gambia is most probably the result of very high mosquito dispersal ranges in that setting, which cause effects to be diluted out through mixing of mosquito populations across treatment arm villages, as demonstrated in Hawley et al 2003 among others and discussed in Killeen et al 2003 Lancet Infect Dis 3: 297. Again, looks intended to create the impression of a long unsolved mystery that is

misrepresentative and exaggerates the novelty and usefulness of this piece.

6. Similarly, in the discussion, outline the role of the insecticide in delivering on the impact of LLINs is not as novel as portrayed and these issues, as well as the effect of resistance and potential impact of resistance management options have been discussed at length in other pieces in authoritative journals in recent years (Briet et al 2013 Malar J 12: 77, Viana et al 2016 PNAS 113: 8975, Killeen 2020 and refs therein). In particular, much bigger picture strategic issues over the short and long term have been explored in a more fundamental way in Killeen 2020). While I certainly agree with the importance of countering the recently initiated and very silly "conversation on the necessity for insecticide in nets" mentioned on line 287, and welcome this contribution to the evidence base that does so, this paper would be much stronger if it presented itself as such, rather than an entirely unprecedented collection of new insights.

MINOR COMMENTS

7. In the opening of the abstract and introduction, the full name for LLINs should exactly match the acronym. If necessary, say "Long-lasting insecticide-treated mosquito nets, or more simply, long lasting insecticidal nets (LLINs)..."

8. Line 44: "never-the-less" should be a single, unbroken word.

REVIEWER COMMENTS

Reviewer #1 (Remarks to the Author):

The manuscript attempts to unite modelling / theoretical work with data analysis from DHS datasets to describe and quantify and compartmentalize the various elements of the 'community' effect of ITNs/LLINs for malaria. This is an ambitious work, partly because while the idea of a community effect is widely discussed, it is understood (or mis-understood), measured and discussed in varied and inconsistent ways throughout the malaria literature.

Thank you for the summary of our manuscript and literature in this field.

The work appears to generally be sensible and the approach to doing reasonable especially on the modeling side. Though the analysis and the role of the DHS/observational data in supporting this work seems less clear, and not well connected to the large amounts literature on the effect of ITNs using real world data.

We thank you for your comments about how to improve the DHS component of our analysis and address your comments in turn below.

We have also more thoroughly discussed the present work in the context of what is known on mosquito net efficacy in the literature. We add to the discussion noting:

- The probable seasonal use of nets which may skew estimates of impact given these interventions are used more rigorously when risk of transmission is greatest. This could reduce any correlations in net use and protection to users or non-users.
- The differences in net use across ages or societal groups.
- The difficulty to improve net access and use further particularly as usage reaches relatively high levels (of 60% use and upward)
- Evidence from alternative sources including experimental huts and randomised control trials.

On L324-351 of the clean manuscript we now write:

“There is no direct method of exploring how reliable our mechanistic modelling results are so we use DHS data to explore how well it can reflect differences in the RDT prevalence of malaria between those individuals who reported they did or did not sleep beneath nets the previous night. Taking the lowest possible spatial scale (clusters), we can see that the broad qualitative conclusions identified by the mechanistic model are statistically supported by the observed data. The model predicts that LLIN users on average have a lower malaria parasite prevalence than non-LLIN users within the same clusters, but this difference is relatively modest, only becoming substantial when approximately half the people in the cluster are RDT positive. Similar trends were seen in the observed data, with only a significant difference between users and non-users in clusters with intermediate disease endemicity. This result is also consistent with recent studies that saw similar patterns between users and non-users [42–44]. Both the mechanistic model and the observed data suggested that the level of net use does not influence the difference between users and non-users. Similarly, the model predicted very little difference between users due to the presence of insecticide resistance and we could not see a change in prevalence of the two groups over time or with increasing pyrethroid resistance in local mosquitoes in the empirical data. The similarities of the model and the observed data afford some comfort that the model assumptions are

appropriate. Nevertheless, care should be taken interpreting results as we are not able to directly measure the different benefits of LLINs, and the deterministic model we employ tends to overestimate the difference in prevalence in LLIN users and non-LLIN users relative to the DHS data. This difference is likely due to the variability in real world data that can stem from multiple sources, and we do not calibrate the model simulations to each cluster individually. For example, the local ecologies of each cluster will likely alter the potential for transmission given ratios of mosquitoes to people associated with rainfall, temperature, and land-use [45–47], the preference of local mosquitoes to bite during the night, on people or indoors [48–51]. The use of nets over seasons may well cycle with peaks in net use occurring when the risk of transmission is greatest [52], and net use may also vary among community members of different ages [53]. Coupled with these socioecological reasons, the addition and use of complementary interventions has altered over time which is likely to contribute further to variability in these DHS data.”

Overall i think the manuscript is important and original and highlights some very important areas in which the community effect of ITNs is misunderstood/poorly quantified and could be helpful for malaria epidemiologists, and modelers health economists and in strategic planning for malaria control programs.

Thank you

Major concerns:

1) The methods for the analysis of DHS data are not clearly reported, and some descriptions of these data are not correct:

We thank you for identifying mistakes in our DHS descriptions. We have corrected them as follows below:

1a) in DHS data a 'cluster' is typically an census enumeration area, which may be as small as a city (or a few) block or part of a village, and has nothing to do with what health facilities are used for treatment (line 355).

Thanks for this, we have changed the text on L452-454 to read:

“In these DHS data, a cluster typically represent a census enumeration area, which can be villages in rural areas or city blocks in urban areas, and represents as close to a homogenous area as possible.”

1b) net usage is not ascertained from head of household. In the current versions of DHS questionnaires this IS part of the household questionnaire, but this questionnaire can be answered b anyone of the age 15 or older in the household. The ways and from whom these questions were asked have changed over time.

Thanks for sharing this. We have adjusted our explanation following the guidance of the reviewer. Since we have included DHS from multiple countries between 2010 and 2018, the text on L455-458 now reads:

“Net usage is estimated from the household survey by asking a household member of age 15 years or older, who in the household slept beneath a LLIN the previous night. DHS surveys have evolved over time and this question has been addressed to different household members.”

1c) the question is not asked by person it is part of the net roster in which it is asked who on the household roster slept under a specific net the night before the survey.

We have addressed this as shown in comment to 1b.

1d) on line 355 a data aggregation to admin 1 is described. Here admin 1 is described as district or province, but admin 1 is generally province or region, district is typically and admin 2 disaggregation.

We have changed the text on L477-479 to say:

“For this linear model, we used a simplified model with the difference in prevalence between non-users and users as the outcome variable aggregated to the first administrative level (often the District or Province level of a country). ”

1c) DHS data follows a complex multistage sampling and stratification and aggregation of data to levels for which the sample was not designed to represent can lead to misleading conclusions. It is not clear why the authors did not use DHS stratifications in this component of the analysis.

We did our analysis at cluster level because net usage and malaria prevalence are not homogenous at the aggregated levels. We did our analysis at the cluster level because clusters are as homogeneous as possible and better reflect similar areas in the model. We have adjusted the text on L452-454 to read:

“In these DHS data, a cluster typically represent a census enumeration area, which can be villages in rural areas or city blocks in urban areas, and represents as close to a homogenous area as possible.”

1d) line 362, what trends are the authors referring in the linear regression? net use? prevalence? resistance predicted by the other? it is not clear from the methods what exactly this analysis is?

We have edited the text to make our methods clearer. From parallel work, we have seen that pyrethroid resistance as measured using a susceptibility bioassay has generally increased over time, and particularly across the years for which we extracted DHS data (2010-2018). This is also a trend identified by other researchers, for instance Hancock et al. (2021). Therefore, we used time as a proxy for pyrethroid resistance, as well as pyrethroid resistance values from a review to see if there was an association between the difference in malaria parasite prevalence of users and non-users (as estimated from the DHS records) over time, including a random effect for country in the regression. The text on L471-489 now reads:

“Pyrethroid resistance, as approximated using susceptibility bioassays, has increased across many areas of sub-Saharan Africa over time [74]. Evidence is building that suggests the presence of mosquitoes with resistance to pyrethroid insecticides may reduce the efficacy of mosquito nets [28, 30, 31, 75, 76].

Therefore, we wanted to explore whether we could see an effect of pyrethroid resistance on the difference between malaria parasite prevalence in net users and non-users. We used a systematic review of susceptibility bioassays collated from 2000 – 2018 across the African continent [77, 78] to investigate changes in resistance. For this linear model, we used a simplified model with the difference in prevalence between non-users and users as the outcome variable aggregated to the first administrative level (often the District or Province level of a country). However, we used pyrethroid resistance (approximated as the proportion of mosquitoes surviving exposure at susceptibility bioassay testing) instead as the predictor variable. Our data consisted of 357 observations across 16 countries. We were able to estimate this predictor of resistance at the administrative 1 level by fitting logistic functions to the available data from each country (Lambert et al. in prep).

Clearly, there are limitations with this broad-brush approach, not least given the variability in both the assay and geographic expressions of resistance [56, 79, 80] as well as varying prevalence over time and space – we present these limitations in the discussion – but we include the analysis to see whether there is a signal in the empirical data, and to give some context for our subsequent modelling exploration of what increasing resistance may mean for continued protection offered by a community effect
”

1e) given the restriction of having at least 5 children using and 5 not using nets in each cluster of inclusion, surely the distribution of clusters is biased away from extreme use/not use clusters in this analysis as well as from small clusters.

We thank you for this comment and have changed our criteria to having more than 20 children to reduce the bias away from extreme use/not use clusters.

2) The comparison between modelled data and DHS data seems to be completely qualitative. This is probably/possibly all that is doable in this particular framework, but the methods don't discuss any approach for this comparison or how the conclusion that the data supports the authors hypothesis.

We thank the reviewer for this observation and have added extra statistical analysis as described below.

2a) line 97 - "the hypothesis that any difference in malaria burden in users and non-users is not influenced by usage" Is a very unclear sentence. As written it seems tautological, since the difference between groups created by conditioning on usage, must be due to usage? I think the authors mean that the absolute difference in prevalence between the groups of users vs. non-users is not influenced by community levels of usage? If so there are straightforward statistical methods for making such assessments such as the testing of interaction terms. I'm a bit surprised that the authors have not chosen to undertake such an approach, or really applied statistical methods to this data at all.

We thank you for this comment about including more statistical methods which we believe has greatly strengthen our paper. Instead of our old model just looking at time, we have now analysed a linear mixed effects model where we include cluster usage, prevalence, an interaction between those two terms and time. We describe our new methods on L461-470:

“We developed a linear mixed effects model from the R package lme4 [73] to investigate what was driving the difference between prevalence in non-users and users (outcome variable). Following visual inspection of the mechanistic model outputs, clusters were grouped into low (0-33%), medium (33-66%) and high (66-100%) malaria prevalence to allow overall trends to be identified. In the main text net usage within the cluster was restricted to low (20-50%) and high (50-80%) net usage to reflect the ranges normally seen in country programmes. Alongside cluster usage and prevalence group we included an interaction between the prevalence and usage grouping and a country specific random effect. Uncertainty was calculated using the bootMer package where 1000 bootstrap samples were taken for each data point.”

And results on L146-161:

“To statistically explore whether our model projections reflect the empirical evidence from the DHS data we conducted mixed-effect logistic regression. Clusters were grouped into low (0-33%), medium (33-66%) and high (66-100%) malaria prevalence and low (20-50%) and high (50-80%) net usage. We see that only the medium malaria endemicity group has a significant difference between users and non-users (median of 3.2% higher, $p = 0.017$). Figure S3 shows the raw data binned in these groups whilst Figure S4 shows the modelled predictions for the difference between users and non-users. Neither year of survey or cluster net usage significantly improves model predictions of the difference in prevalence between non-users and users. This supports the hypothesis generated by the mechanistic model that the absolute difference in prevalence between users and non-users is not influenced by community levels of net usage but more associated with the local endemicity of a setting. The interaction term between prevalence and usage categories is insignificant ($p = 0.197$ and 0.499), suggesting the difference in malaria prevalence between users and non-users is also consistent. Overall, there is less of a difference in the field data between users and non-users for a given prevalence compared to the model suggesting current model structural assumptions may be exaggerating the difference. Analyses across all net usage levels is shown in the supporting information.”

2b) line 104 - similarly the authors state that they did not "detect a signal" that the difference in prevalence between users and non-users changed over time, or with level of resistance. but don't clearly describe the method used to search for this signal. I believe that this is a linear regression aggregated to the admin one level, for resistance, though for change over time it is less clear. Regardless neither the outcomes nor the exposure variables, the dataset size etc... are fully described in the methods for this.

We thank you for this comment and your assumption about the original methods was correct. We have however adjusted this analysis based on your other comments above as described.

3) the authors focus very much on presenting results as reductions in EIR to users vs. non-users and throughout don't present overall reductions in EIR consistently. While the disaggregated approach is helpful for highlighting the difference, it also sort of masks the overall meaning of this effect on the outcomes which are of the most clinical and public health meaning, which are reductions in EIR at the community level or reductions in parasite burden at the community level. because the importance of the protection of users vs. non-users changes and the prevalence of use changes, its important to think about community protection too.

We agree with your comment. In the analysis we were originally aiming to understand the components of protection, explicitly to different groups of the whole community, that is the net

users and non-users. Therefore, we wanted to distinguish the protection afforded to each. However, we agree that it is worth also presenting the reductions in EIR that are afforded to the total community as this is the critical aim of any population level intervention. We have added this estimate to the results in Figures 3, 4 and in the SM along with the following textual edits:

L 201-204:

“We can also summarise the direct benefit of the barrier at the community level. When one person in the community uses a net, the relative reduction in EIR is minimal (0 [95% CI: 0 - 1]). However, when 80% of the community are using nets, there is a relative reduction in EIR of 11 [95% CI: 5 - 25] to the community.”

L 210-212:

“When 80% of the community are using nets, we estimate a relative reduction in EIR of 24% [95% CI: 5% - 41%] for users, 12% [95% CI: -3% - 40%] for non-users and 22% [95% CI: 4% - 41%] for the community (Figure 3B).”

L 230-232:

“At the community level, again there is no meaningful reduction when just one person uses a net but the relative reduction when 80% of the population use a net is 36% [95% CI: 8% - 65%].”

L 240-242:

“The same pattern occurs at the community level with negligible relative reduction in EIR for one individual using a LLIN but 14% [95% CI: 5% - 26%] when 80% of people are using them.”

L 251-255:

“This is minimal when only one user in the population has a net (reduction in EIR for both users, non-users and community is 0% [95% CI: 0% - 1%]) (Figure 3D), but further reduces the EIR by 30% [95% CI: 7% - 44%] in users, 58% [95% CI: 45% - 76%] in non-users and 37% [95% CI: 16% - 48%] at the community level when 80% of the population sleeps under a net.”

L 263--267:

When usage increases to 80%, the reduction in EIR for LLINs is 89% [95% CI: 67% - 98%] for users, 74% [95% CI: 48% - 92%] for non-users and 87% [95% CI: 63% - 96%] for the community (Figure 3E). In contrast, the total protection offered by an untreated net used by 80% of the population is 42% [95% CI: 11% - 71%] reduction in EIR for users, 12% [95% CI: -3% to 40%] for non-users and 36% [95% CI: 8% - 65%] for the community

4) a sort of follow on to number 3 is that the authors suggest at least twice in the manuscript that the implications of the analysis are that more protection is always favored. This is only true when one considers 'more protection' as unconstrained maximization of health benefits with no cost function or alternative. This is a view that is, at least I think, misleadingly simple. The authors have done the hard work of essentially estimating the effects of varying coverage, including the mass effects of nets, but completely neglect to describe or consider the marginal gains made by increasing coverage which can be estimated from the outputs here (or to even mention that such gains inevitably come at a cost which will be a tradeoff against something).

We agree but we wanted to stress that from a public health perspective, the more people who use nets the greater the level of efficacy and the greater the reduction in EIR and malaria parasite transmission. We were nervous to have the work mis-interpreted, which has happened previously for studies on a similar theme. That is, we see the maximum difference in prevalence for users compared to non-users at around 60% use. This does not mean that 60% net use is the best public health target. We now add to the discussion to align this work more appropriately with previous literature. In doing so, we discuss the recent paper by Bertolli-Villa et al 2021 that shows the inevitable trade-off in cost-effectiveness and increased accessibility to nets mentioned. We have added on L393-406:

“Results here and elsewhere [9, 19, 25, 61] illustrate the importance of achieving high coverage with both LLINs and untreated nets as increasing net use generally will be beneficial for community protection (Figure S7). Killeen [19] argues that prioritising vector mortality rather than universal usage may be more cost-effective if newer products, with alternative active ingredients, are more expensive (and therefore fewer can be procured for a given region). This makes sense if the killing effect is particularly potent, and nets are distributed in a way that means non-users are in close enough proximity to benefit from the reduced numbers of mosquitoes caused by the killing action of the net. Novel net designs that either chemically [62–64] or mechanically [65] kill mosquitoes are welcomed in order to mitigate for the potential loss in impact from pyrethroid-LLINs [31, 66]. It may be cost-effective to then top-up communities with untreated nets to provide some barrier protection to those members who do not have the treated net. A cost-effective analysis is beyond the scope of the current work, particularly because it becomes logistically harder and more expensive to increase usage once net use has reached around 60% so the association is non-linear [67], but certainly worth consideration for future discussion.”

4a) I wondered also why in the interest of symmetry the authors didn't model an even higher coverage scenario, such as nearing 100% usage of nets. While these are unlikely to be achieved in real life, this might help fill the whole spectrum of net usage. Despite the analysis restrictions imposed by the inclusion criteria the authors have chosen to model a scenario at the low usage end which cannot be found in real world data. One wonders why not one at the high end.

We thank you for this suggestion but feel that for our scenario figures (Fig 3, 4) we wanted to only select a few achievable coverages to reduce the complexity of the plot. Instead, we have added an additional figure (Figure S7) to address point 4 where we include 100% usage.

5) On line 279-280: the authors make a statement "there is no minimum LLIN usage needed for users and non-users to benefit from community protection" This statement, which i think the paper supports, and I also think is true is a point which is missed by a huge proportion of the community and might deserve more emphasis. The idea that there is some threshold coverage that must be achieved for community protection is widely held, and the idea that community protection benefits not only non-users but also other users is also often seems to be missed. the authors might make this a clear point of their findings

We agree this is an important finding and have added it to the third paragraph of the abstract on L31-35 which now reads:

“We find that there is no minimum LLIN usage needed for users and non-users to benefit from community protection. Modelling results indicate that pyrethroid resistance in local mosquitoes will likely diminish the direct and indirect benefits from insecticides, leaving the barrier effect for net-users intact, but LLINs are still expected to provide enhanced benefit over untreated nets even at high levels of pyrethroid resistance.”

More minor comments:

1) throughout the main text and supplementary information the tables and figures sometimes lack complete notation descriptions making them difficult to interpret as stand alone documents: examples include in the supplementary information, the human compartment model diagram, P is not described nor are any of the flow parameters, in the first table illustrating the comparisons the a, b, c, d, and control notation is used, but without a caption/note to help the reader understand these. Which requires cross reference to text which may not be easy to find in final production. Etc...

We thank you for this comment and have gone through our table and figure captions and added extra explanation, especially in the supplementary text.

2) line 224- "the difference is because of the innate variability in real world data" doesn't mean much.

Apologies, we have expanded this on L342-351 to be:

“This difference is likely due to the variability in real world data that can stem from multiple sources, and we do not calibrate the model simulations to each cluster individually. For example, the local ecologies of each cluster will likely alter the potential for transmission given ratios of mosquitoes to people associated with rainfall, temperature, and land-use [45–47], the preference of local mosquitoes to bite during the night, on people or indoors [48–51]. The use of nets over seasons may well cycle with peaks in net use occurring when the risk of transmission is greatest [52], and net use may also vary among community members of different ages [53]. Coupled with these socioecological reasons, the addition and use of complementary interventions has altered over time which is likely to contribute further to variability in these DHS data.”

line 228 - what is meant by 'sampling bias' and how is the exacerbating variation? Bias generally means bias, and is independent of sample size. Variance or variation, while also independent of sample size directly integrates into precision which is affected by sample size? also this is an incomplete sentence.

We thank you for this comment. We have re-written our discussion in line with comments from reviewer 3 and no longer include this sentence.

line 271-273 : increasing net use always increases protection... again ignores the question of whether or not this increase is meaningful. It a pretty limited statement to note that the relationship between net use and impact is monotonic, describing the shape of this function (and its first derivative) would be of use especially in priority setting.

Similar to our response to question 4, we agree that this function is non-linear, but we are nervous to suggest some threshold estimate that might be easy to misinterpret. Having considered the reviewers points here and on the trade-offs in cost-effectiveness and increasing net use, we have added an additional Figure S7 to our results with the following description in L269-270:

“However, this relationship is non-linear (Figure S7), with lower usages providing a larger cumulative benefit than higher usages.”

and added the following to the discussion on L393-406:

“Results here and elsewhere [9, 19, 25, 61] illustrate the importance of achieving high coverage with both LLINs and untreated nets as increasing net use generally will be beneficial for community protection (Figure S7). Killeen [19] argues that prioritising vector mortality rather than universal usage may be more cost-effective if newer products, with alternative active ingredients, are more expensive (and therefore fewer can be procured for a given region). This makes sense if the killing effect is particularly potent, and nets are distributed in a way that means non-users are in close enough proximity to benefit from the reduced numbers of mosquitoes caused by the killing action of the net. Novel net designs that either chemically [62–64] or mechanically [65] kill mosquitoes are welcomed in order to mitigate for the potential loss in impact from pyrethroid-LLINs [31, 66]. It may be cost-effective to then top-up communities with untreated nets to provide some barrier protection to those members who do not have the treated net. A cost-effective analysis is beyond the scope of the current work, particularly because it becomes logistically harder and more expensive to increase usage once net use has reached around 60% so the association is non-linear [67], but certainly worth consideration for future discussion.”

a minor technical point I think that Insecticide resistance is parameterized throughout as simply decreased killing of mosquitoes, but in practice there may also be changes to the repellency/deterrence. It may be worth noting that this may not fully capture the nuances of IR.

We thank you for this comment. From the data perspective, we define pyrethroid resistance as the proportion of mosquitoes that survive exposure to a discriminatory dose of pyrethroid insecticide in the susceptibility bioassay. To parameterise the transmission model however, we use this metric to define how the presence of an ITN alter the outcome of a mosquito feeding attempt. This does impact on mosquito mortality, repellence, successful feeding and deterrence as the reviewer rightly notes. (This statistical work is published in Nash et al. 2021 and Sherrard-Smith et al. 2022 and validated in Sherrard-Smith et al. in press.)

We now clarify this in the supplementary methods (Text S1):

“The proportion repeating (k_{N0}), feeding successfully (s_{N0}) and dying (d_{N0}) are altered in a setting where local mosquitoes show resistance to pyrethroids. In previous work, we use meta-analyses to estimate the association between the probability of mosquitoes surviving exposure to a discriminatory dose of pyrethroid in the susceptibility bioassay, and the mortality observed in experimental hut trials testing the entomological impact of mosquito nets. We also learn the associations between the outcome of a mosquito feeding attempt in the experimental hut linking mortality to successful feeding, repellence and deterrence (ref). We have then estimated the additional benefit from pyrethroid-

PBO nets so that we can explore the change in r_{NO} , s_{NO} and d_{NO} with different measures of susceptibility bioassay survival (ref). We have validated this process using a systematic review of randomised control trials as the gold standard method to estimate epidemiological impact of interventions (ref). We use these estimates to explore how resistance impacts on the barrier and community effect provided by the mosquito nets.”

and adjust the caption for Table S1:

“**Table S1. Uncertainty parameter ranges used to explore model variation.** The untreated and treated net (LLIN) parameters are shown for specified levels of resistance where 0 indicates the absence of pyrethroid resistance (100% of mosquitoes are killed on exposure to a discriminatory dose of pyrethroid insecticide during bioassay testing), 40 and 80 indicate 60% and 20% of mosquitoes are killed on exposure respectively during discriminatory dose bioassay testing. Repellency of mosquitoes from LLINs is also impacted by resistance and is captured by the r_{NO} parameter.”

line 290 and 291: its a bit confusing to say maximize the use of LLIN then increase coverage with untreated nets. I think in practice it may actually be relevant, since there are populations that may never get/use government issued LLIN but could procure untreated nets on the market (such as in urban areas) but this reads as if you are advocating for 100% usage of LLIN then adding untreated nets.

Thank you, we have added a new Figure S7 and edited the text on L393-406 to read:

“Results here and elsewhere [9, 19, 25, 61] illustrate the importance of achieving high coverage with both LLINs and untreated nets as increasing net use generally will be beneficial for community protection (Figure S7). Killeen [19] argues that prioritising vector mortality rather than universal usage may be more cost-effective if newer products, with alternative active ingredients, are more expensive (and therefore fewer can be procured for a given region). This makes sense if the killing effect is particularly potent, and nets are distributed in a way that means non-users are in close enough proximity to benefit from the reduced numbers of mosquitoes caused by the killing action of the net. Novel net designs that either chemically [62–64] or mechanically [65] kill mosquitoes are welcomed in order to mitigate for the potential loss in impact from pyrethroid-LLINs [31, 66]. It may be cost-effective to then top-up communities with untreated nets to provide some barrier protection to those members who do not have the treated net. A cost-effective analysis is beyond the scope of the current work, particularly because it becomes logistically harder and more expensive to increase usage once net use has reached around 60% so the association is non-linear [67], but certainly worth consideration for future discussion.”

line 306-307: inferring the level of insecticide resistance reduces the 'power' of the analysis? How? This seems an odd conclusion. Also how do you know that the dataset is underpowered to detect the clinical differences suggest by the model, have you done a formal power analysis to justify this?

We did not do any formal power analysis so have removed this. The sentence on L418-420 now reads

“The level of resistance in each DHS cluster was not directly measured but instead inferred, reducing the strength of the analyses.”

Reviewer #2 (Remarks to the Author):

The authors explore an interesting question of attributing the individual and community effects of bednets, further distinguishing the effects of the bednet barrier vs insecticide, and consider the impact of both bednet coverage and insecticide resistance on modulating these effects. They use a deterministic modeling approach with parameterization based on statistical models fitted to experimental hut trial data. The authors also compare prevalence in net users and non-users at the cluster level using DHS surveys across multiple countries, years, and epidemiological contexts.

Thank you for this summary of our research.

Gathering quantitative evidence of the individual- and community-level effect of ITNs is important and helpful for decision-makers, especially in the context of insecticide resistance and continued need for effective vector control. However, I felt that the work as presented was a missed opportunity to use a modeling platform to more deeply explore questions around ITN effect size that are extremely challenging to measure in the field. Furthermore, several aspects of the presentation were confusing and would benefit from clarification.

We decided to use this type of model because it gave us the ability to tease apart the different parts of protection offered by the net more easily, despite its limitations in exactly matching the DHS setting. We address your comments below.

Major comments:

1. It is unclear what to make of seeing both the DHS and modeling results together. They don't seem to be in agreement beyond both seeing the same direction of effect (a low bar). The authors list reasons why this could be, but choose not to investigate any of them (perhaps because of major comment #2 below). At the end of the day, I'm not sure why I should believe any of the model results are relevant to the real world since they are not reconciled to actual measurements. It would still be interesting to look at this question of ITN effect size attribution in an abstracted model, but the authors' framing is very much toward quantitatively assigning these attributions rather than a general exploration of the relative magnitude of effects.

We thank you for your comments about our study design. As suggested by reviewer 1, we have enhanced our DHS analysis including statistical tests to support our claims about trends. We have taken on board your comment about us using an abstracted model and have made this clearer in the:

Results L173 of the tracked manuscript:

"Results are presented for a hypothetical, non-seasonal setting."

Discussion L380-383:

"Nevertheless, we estimate the relative protection provided by the insecticide is considerable, the relative reduction in EIR for LLINs is 89% [95% CI: 67% - 98%] for users and 74% [95% CI: 48% - 92%] for non-users in areas in our hypothetical setting (simulating an EIR of 100 prior to the introduction of 80% net use)."

Methods on L578-580:

"We illustrate the impact of the mass community effect by investigating the reduction in both EIR and all-age prevalence from the control scenario for users and

non-users for hypothetical non-seasonal settings with pre-intervention EIRs of 100 and 10.”

2. It seems that choosing to use a deterministic compartmental model rather than an individual-based model has really limited the authors’ ability to investigate their research question in more detail. For example, imperfect compliance to net usage (individuals occasionally do not use their net), heterogeneity in exposure (correlated, uncorrelated, or anti-correlated with net usage), and spatial effects could each contribute to the attenuated difference observed in DHS vs in the model. The authors could potentially evaluate the possible contribution of these types of secondary (or primary?) effects in a modeling framework, and this work would be both more interesting and more relevant.

We chose to use a deterministic model rather than an individual based model because we could easily tease apart the different parts of protection offered by bed-nets (the main aim of our analysis), which would be much harder with the individual based model. We agree with the reviewer, and have already written in our limitations, that with an individual based model we would be able to match the settings of the DHS data more closely. However, there are still important parameters such as previous bed-net usage that we still would not be able to get from the data. DHS data do not record seasonal patterns in net use so it would not be possible to use these data to explore imperfect compliance. A spatial model would be a most interesting progression given that we cannot account for closeness of net users and non-users with the current framework, but this is beyond the scope of our current analysis. We feel at this stage it is out of scope to switch to an individual based model but will consider the suggested analysis going forward.

Minor comments:

1. Please add page numbers, this will make it easier for reviewers as we flip back and forth.

We have included page numbers and reference those throughout our response to reviewers, apologies.

2. L67-68: this is confusing. What is “this protection” that L67 is referring to? The barrier? The net in general? The insecticide? Are you then saying that to the individual-level protection there is added i) physical barrier protection, which I would think would already be included in individual-level protection?

Thanks for highlighting this confusing sentence. We have re-written the introduction following suggestions from reviewer 3 and no longer include this sentence.

3. L97: “the trends match” could you elaborate on this and explain exactly what is meant?

The trends in the absolute difference in prevalence between users and non-users. As suggested by reviewer 1, we have used a linear mixed effects model to describe the relationship between usage, prevalence and the difference in non-user and user prevalence.

We describe our new methods on L462-470:

“We developed a linear mixed effects model from the R package lme4 [73] to investigate what was driving the difference between prevalence in non-users and users (outcome variable). Following visual inspection of the mechanistic model outputs, clusters were grouped into low (0-33%), medium (33-66%) and high (66-100%) malaria prevalence to allow

overall trends to be identified. In the main text net usage within the cluster was restricted to low (20-50%) and high (50-80%) net usage to reflect the ranges normally seen in country programmes. Alongside cluster usage and prevalence group we included an interaction between the prevalence and usage grouping and a country specific random effect. Uncertainty was calculated using the bootMer package where 1000 bootstrap samples were taken for each data point.”

We describe our new results on L146-161:

“To statistically explore whether our model projections reflect the empirical evidence from the DHS data we conducted mixed-effect logistic regression. Clusters were grouped into low (0-33%), medium (33-66%) and high (66-100%) malaria prevalence and low (20-50%) and high (50-80%) net usage. We see that only the medium malaria endemicity group has a significant difference between users and non-users (median of 3.2% higher, $p = 0.017$). Figure S3 shows the raw data binned in these groups whilst Figure S4 shows the modelled predictions for the difference between users and non-users. Neither year of survey or cluster net usage significantly improves model predictions of the difference in prevalence between non-users and users. This supports the hypothesis generated by the mechanistic model that the absolute difference in prevalence between users and non-users is not influenced by community levels of net usage but more associated with the local endemicity of a setting. The interaction term between prevalence and usage categories is insignificant ($p = 0.197$ and 0.499), suggesting the difference in malaria prevalence between users and non-users is also consistent. Overall, there is less of a difference in the field data between users and non-users for a given prevalence compared to the model suggesting current model structural assumptions may be exaggerating the difference. Analyses across all net usage levels is shown in the supporting information.”

4. L111: what is meant by “in the absence of data”? what kind of data?

We are sorry for the lack of clarity. We have re-written that part of the results and removed this sentence.

5. L121: clarify that in this artificial scenario, you enforce that EIR on the non-users is unchanged. I found the framing of the “direct benefit” scenarios (EIR on users is allowed to respond to the presence of the net, but EIR on non-users is forced to maintain original value) unclear on first read-through of this section. I figured it out after digging through the methods, but it would be helpful to have an explanation of this approach to EIR also stated in the results. I found the term “fixed EIR” confusing since EIR is only fixed on the non-user population, and “varying EIR” confusing since I would interpret that as “seasonally-varying”.

Thanks for this suggestion, we have clarified this sentence on L178-180 as follows:

“With our definition the barrier does not influence non-users, regardless of usage, so EIR remains unchanged because the number of infectious mosquitoes and the biting rate of non-users has been set to equal the control scenario. This is fixed throughout the simulation as we present a non-seasonal scenario.”

6. L129-133 and elsewhere: I found the concept of “there are more bites because a vector deterred by a barrier will attempt to bite again, and the more net-users there are, the more this happens” to be confusing. Are the total number of bites not the same if there are nets vs no nets (each vector

feeds at most once?)? Is this a real phenomenon or an artefact of the model? What happens if you change the model so that this doesn't happen? It feels like there's a lot of airtime given to this concept, possibly more than it deserves given that we don't know if it's real (?) AND that the direct protection from the barrier is stronger than the increase in biting.

This is the result of the necessary modelling steps to separate out the direct benefit of the barrier where we fixed the biting rate for non-users. We have added in this explanation on L194-196:

“As usage increases, we propose more mosquitoes will attempt to bite on users that have previously tried to bite on other users (as the biting rate for non-users has been fixed for this scenario).”

7. L138: “more mosquitoes are killed” is this a meaningful increase for the barrier-only bednet?

In our understanding and model framework, this does apply to barrier only nets because of the extended time required for a mosquito to find a blood meal. This increases her risk of mortality. We have clarified this on L212-216 as:

“As usage increases, in general, more mosquitoes die, so the median reduction in EIR for each non-user increases. The reason for this is that the transmission model assumes that a mosquito unable to feed due to the action of the barrier results in an extended foraging time required to successfully blood feed which leads to increased risk of mosquito mortality (per successful blood-feed).”

8. L174: “to a single person” unclear if this means per individual or if it means in the situation where only 1 person is using a net

This refers to only one person using a net. We have clarified this on: L258-260

“The overall protection provided to one individual using an LLIN is estimated to be a relative reduction in EIR of 62% [95% CI: 36% - 74%] (sum of the above four different types of protection, Figure 3E).”

9. L192: any key takeaways from the analysis on reduction in all-age prevalence?

We thank you for this missed opportunity. We have added an extra sentence on L281-282:

“The patterns seen for changes in EIR are similar to those found for prevalence.”

10. L197: I thought repellency was part of the barrier model, does pyrethroid resistance not impact repellency?

In our model assumptions, untreated nets repel mosquitoes physically. This is distinct to the potential additional repellence induced by the insecticide in treated nets. The combined impact is estimated from a meta-analysis of experimental hut data in Nash et al. (2021).

11. L201 and others: what is a reduction in EIR of 22? Fig 4 caption says initial EIR = 10, how could it reduce by 22? Or is this 22%?

Thanks for catching this mistake. Figure 4 is for the scenario with an initial EIR of 100. The figures for the analysis repeated with an initial EIR in 10 are in the supplementary material.

We have rewritten the section from the results on L291-305 more clearly to specify that we are referring to a relative reduction of 22 bites per person per year and throughout this section.

“For example, in the scenario shown in Figure 4 we start with an EIR of 100 infectious bites per person per year with no pyrethroid resistance in the local mosquito population and 50% of people are using nets. The additional direct benefit of the insecticide for a LLIN-user is a relative reduction in EIR of 22% [95% CI: 9% - 39%] bites per person per year, but this falls to only an relative reduction of 15% [95% CI: 6% - 26%] bites per person per year when the local mosquito population exhibits 80% levels of pyrethroid resistance (as defined by survival of a discriminating dose bioassay, Figure 4A). In addition, at the same usage level, the additional indirect benefit of insecticide is a relative reduction in EIR of 21% [95% CI: 6% - 27%] bites per person per year in the absence of pyrethroid resistant mosquitoes for net users, 46% [95% CI: 31% - 55%] for non-users and 33% [95% CI: 26% - 37%] for the whole community. This falls to a relative reduction in EIR of 15% [95% CI: 9% - 17%], 26 [95% CI: 16 - 35] and 21% [95% CI: 15% - 24%] bites per person per year respectively at 80% pyrethroid resistance in local mosquitoes (Figure 4B). Overall, at 50% net usage the insecticide causes 68% [95% CI: 46% - 89%] of the protection provided by LLINs, with this value reducing to 62% [95% CI: 42% - 82%] in areas with 80% resistance. Similar patterns are seen between users and non-users for different levels of LLIN usage.”

12. L207: “greater reductions” reductions in what?

Absolute reductions in EIR from the equilibrium starting scenario of 100 bites per person per year. We have endeavoured to clarify this entire section (please see point 11). This has been added to the text on L306-307:

“Overall, the relationship between the level of resistance and protection is non-linear, with greater absolute reductions in EIR seen at higher levels of resistance in the local mosquito population.”

13. L213-4 and L338-9: “the role that LLINs play in reducing mosquito populations cannot be underestimated” what does this mean? If the role cannot be underestimated, does that mean that it’s very low to begin with? Or did you mean overestimated? Of course the role can be overestimated, we still have malaria. Not sure what’s going on here.

We choose this phraseology here because literature e.g. [10, 11] has suggested that due to resistance untreated nets should be used. We wish to emphasise that there is additional benefit of using insecticide treated nets even when mosquitoes demonstrate some resistance. We have removed this terminology in our re-write.

14. L233: DHS will not report on variability in night-by-night usage but surely there are research studies that have looked at this. In fact some studies are cited a few sentences later, and general assumptions could be made from other data sources.

Yes – we agree estimates could be taken from other literature. However, this won't match the exact setting for each cluster and so would still be an assumption that would lead to inaccuracies in individual based runs. We have added clarification on L358-360:

“There is currently no method of assessing this variability from the DHS data for each cluster which is designed as a cross sectional survey asking whether a child slept under a net the previous night (although other unmatched data exists e.g. [55]). “

15. L265-9: is this consistent with empirical data?

These numbers are for a hypothetical scenario so no empirical data is available. We have adjusted the description on L380-383 to say:

“Nevertheless, we estimate the relative protection provided by the insecticide is considerable, the relative reduction in EIR for LLINs is 89% [95% CI: 67% - 98%] for users and 74% [95% CI: 48% - 92%] for non-users in areas in our hypothetical setting (simulating an EIR of 100 prior to the introduction of 80% net use”

16. L288-9: yes the insecticide is powerful but this sentence implies to me that the barrier has no effect. Perhaps reword

Thanks for the suggestion, we have re-written that part of the discussion and removed this sentence.

17. L307: “under-powered” is this meant in the technical sense of power? It seems not, in which case a different term would be more appropriate

Thanks for this suggestion – we have changed L418-420 to strength:

“The level of resistance in each DHS cluster was not directly measured but instead inferred, reducing the strength of the analyses.”

18. L359: why RDT prevalence and not microscopy?

We chose RDT because microscopy was not always included in the survey e.g. Uganda 2016 only used RDT (<https://dhsprogram.com/pubs/pdf/FR333/FR333.pdf>).

19. Methods: please also describe assumptions on seasonality (no seasonality), mixedness (well-mixed (?), and age effects on biting risk and ITN usage. LLINs are distributed at the beginning of years 1 and 4, is that right (Fig 1A suggests year 4 redistribution)? The year 4 event is not mentioned (L463).

Further description of the model is provided in the supplementary material Text S1, but we have added the requested details to the main body of the text:

L497-498:

“For this study we assume a well-mixed population with no seasonality.”

L499-501:

“The community considered is split into age, heterogeneity and intervention compartments. Each age compartment is assigned a unique biting rate with the parameters that determine the relationship between age (i.e. body size) and biting rate described In Text S1.”

L503:

“We assume a LLIN user always uses their net, with a three-year waning cycle.”

We have also made clear about the net distribution on L584-587:

“Initially, we subtract the average of the EIR or prevalence between years 3 and 6 from the control (to average over the three-year waning cycle; nets given in year 1, 4, 7), depending on the observation so we are presenting a reduction in EIR or prevalence due to the intervention.”

20. L408-17: what are the units on lambda? Is I_M per person, or total over the population? How is lambda related to the r, d, and s parameters? Does a repeated bite attempt mean the mosquito ages a day between attempts or not?

Thanks for raising this. When simplifying equation (1) from equation (41) in Text S1 we had defined our parameters incorrectly. I_M is the number of infectious mosquitoes and lambda is the intervention compartment-dependent rate at which a person is bitten by a mosquito.

The relationship between r (the probability a mosquito is repelled on a single feeding attempt), d (the probability a mosquito dies), s (the probability a mosquito successfully feeds) and lambda is complex but described fully in Text S1 in the Long-lasting insecticidal nets model. The mortality rate is increased if a mosquito encounters insecticide on a net. Without vector controls, the probability of surviving a feeding attempt is modelled as a function of the time spent in search of a blood meal and the mortality rate for a mosquito vector. Vector controls are assumed to reduce the effective biting rate on each human based on their level of protection (provided by the nets). Lambda is expressed in terms of the human blood meal rate, the relative proportion of bites taken on each human, the probability of a successful feeding attempt and the probability of feeding and surviving.

The background mortality of mosquitoes means that the probability of feeding and surviving decreases each feeding attempt as time passes.

21. L464: using “EIR/prevalence” to mean “EIR or prevalence, depending on which outcome we’re observing” is confusing, it looks like you are dividing EIR and prevalence!

Thanks for your suggestion. We have changed the sentence on L584-587to be:

“Initially, we subtract the average of the EIR or prevalence between years 3 and 6 from the control (to average over the three-year waning cycle; nets given in year 1, 4, 7), depending on the observation so we are presenting a reduction in EIR or prevalence due to the intervention”

22. L670 Table 1: should be figure 2 instead of figure 1

Thanks – fixed.

23. L680-3 Fig 1B caption: is the x-axis EIR in the absence of nets? Or EIR in the subpopulation?

This is the initial EIR prior to nets being added. The caption now reads:

“The dependency of all-age malaria parasite prevalence on the annual initial EIR for the baseline pre-intervention scenario (black solid line), the LLIN users (blue dashed line), and non-users (red line). The initial EIR of 100, which is the simulation shown in A, is indicated by the vertical dashed line.”

24. L707 Fig 3, S3, 4: would be easier to read values off these figures if the gridlines were at 10% intervals

Thanks for your suggestion. I have changed this.

25. L710: 5 scenarios, not 8, I believe

Yes correct, thanks – fixed.

26. L707 Fig 3: I recommend splitting this into 5 panels A-E. In the results section you could then refer to specifically which panel you are discussing, which I for one would find quite helpful for navigating this figure.

Thanks for this suggestion. We have changed Fig 3, 4, and the ones in the supplementary material to be panelled and referred to them in the text in the results section.

27. L734 Table S1: the uncertainty ranges on the treated net parameters d_{NO} and r_{NO} seem fairly narrow, especially given that Nash et al. plots show really large variability in feeding success. Also: is net retention not modeled at all? If so, should note in methods.

We thank you for this comment and have added extra explanation into Text S1 to describe our methodology:

“Following the same logic, we use the data from Nash et al. [6] to estimate what these parameters are likely to be in the presence of untreated mosquito nets. There were 90 data for untreated nets within the meta-analysis. The median estimate with the 25th and 75th percentiles for the proportion feeding successfully, being killed or being repelled were used to calculate j_1 (0.55, 0.42 – 0.62), k_1 (0.37, 0.26 – 0.55), and l_1 (0.08, 0.03 – 0.14) respectively.

Combining these provided estimates for the model parameters r_{NO} , s_{NO} and d_{NO} for untreated nets (Table S1.5).”

Net retention is included assuming exponential loss of nets from the community such that after 5 years, half as many people would still be using the net, though we simulate net distributions every 3 years and replace old nets. We assume the same people get given nets each time. We add the following description on L353-355:

“Our model assumes systematic use of LLINs over time, i.e. users will always use a LLIN each night whilst non-users will not, though community use is simulated to wane with time since the previous mass campaign.”

28. L789 Fig S3: a lot of this is the same as Fig 3, right? But the yellow means something different than it did in Fig 3. Perhaps recolor the S3 yellow to something else, I was pretty confused what was different vs not between the 2 figures.

Thanks for this suggestion. We have changed the yellow colour in Fig S6 and Fig S10 to pink.

29. L809 Fig S4: it's not easy to look across these plots and spot differences. Why not pick a few most relevant of the scenario-coverage combinations and plot outputs vs EIR? Would be easier to assess the impact of EIR on effect size.

The new Figure S8 is the same as Figure 3 but for an EIR of 10. We have included this figure to show the same trends exist for a different hypothetical setting.

We have added in an additional Figure S7 where we show how the community protection varies as community usage varies.

Reviewer #3 (Remarks to the Author):

MAJOR COMMENTS

1. In the abstract, the following statement is not entirely true and the quotes should be removed from the community effect, because it's a very real, well established phenomenon: "The inherent linkage between these types of protection renders it impossible to empirically quantify the 'community effect'." For example, see figure 2 in Hawley et al 2003 AJTMH 68(Suppl4): 121 for an example of more-or-less full community effects experienced by people lacking nets but living right beside the boundary with areas enjoying full coverage.

We thank you for this comment. It is our understanding that it is difficult to empirically quantify a community effect. Were we to try to estimate this through gold standard experimental hut, randomised control trials, or other experimental frameworks, we would be unable to isolate the barrier effect of the net for the user i.e. the degree that the barrier acts to isolate the net-user from local mosquito bites.

We can only measure population level effects on various mosquito outcomes that inform transmission metrics like the vectorial capacity. Even with the Hawley et al. (2003) publication, which is an excellent early study supporting the presence of a community effect from ITNs, we cannot isolate the different protections offered to individuals across the population. Our attempt here is to do this so that we can try to understand the different effects that ITNs can have, particularly to try to quantify the barrier effect and then the indirect benefits from the induced killing of mosquitoes that is of course a key action of insecticidal nets.

2. The bottom-line conclusions in the abstract represent nothing new and have been both modelled and demonstrated empirically many times over the years. Indeed, relevant principles (Killeen et al 2007 PLoS Med 4: e229) and supporting evidence (Pryce et al. Cochrane Database of Systematic Reviews 2018, CD000363) were accepted and adopted as WHO policy back in 2007 (<http://medi-guide.meditool.cn/ympdf/FBACA171-AF95-DCFA-673E-3FA3AFEB79AE.pdf>). While the modelling here looks solid and there's nothing incorrect about the conclusions, presenting them as new insights, rather than old but often ignored ones (Killeen 2020 Lancet 395: 1394), is misleading and I'm sure the authors know as much.

Thank you for highlighting these important papers that we had not considered previously – apologies for missing these. We had not previously read the 2020 review but are glad to have done so now. We agree that there are aligned conclusions but explain below some of the differences we see in the approach we present. As a result, we have altered the abstract, introduction and discussion to highlight these distinct findings and to better place the presented analyses in the context of previous works.

The Killeen et al. 2007 publication presents a method to identify the individual protection offered by a mosquito net and the protection to non-users and looks at the change in these estimates as coverage of the community increases. It uses relative exposure as the metric for estimating an effect size for these denominations and considers dipped nets that have more rapid waning of the killing effect from the insecticide and the potential of improved long-lasting insecticidal nets. The paper identifies individual protection and communal protection. In our approach, we consider the barrier effect and the indirect benefit from the insecticide to a net-user, but then separate the community benefit further into the direct and indirect benefits to others not using nets due to the insecticide and collate these as an overall benefit. In our opinion, this is useful particularly in the context of increasing pyrethroid resistance across malaria-endemic countries because we can use the process to quantify how these benefits may be affected. This contributes to the debates ongoing on the

value of ITNs and untreated nets but also generates ideas of how novel interventions could complement the nets as specific elements of protective effects are diminished.

The Pryce et al. 2018 review summarises evidence from individual and cluster-randomised control trials (RCTs) on the overall effect size of treated and untreated nets relative to each other, insecticide-treated curtains, or no intervention. It does not attempt to quantify the individual protection afforded for net-users and the potential additional protection offered to non-users. One challenge for Cochrane reviews is that it is difficult to account for the transient element of the RCTs. That is, some of the trials will measure epidemiological outcomes after 6 months, others after 2 years etc. We know the protective effects of nets diminish with time. Further, we know that ecology alters the optimal impact any intervention can have so comparing multiple studies using the Cochrane review has some limitations. A benefit of a mechanistic model is that once we can validate the outcomes against empirical data, we can also compare like-for-like ecologies and cross-sectional surveys because we can account for the different locations, time series and study designs that are used within different RCTs. Since submitting the present manuscript, we have comprehensively validated our model using a review of randomised control trials (N = 13 that met the inclusion criteria) (Sherrard-Smith et al. in press) which gives us further confidence in the transmission mechanism adopted in the present analysis.

Given this, we have updated the introduction to better represent previous work in this space on L55-69 of the clean manuscript.:

“The challenge to deliver LLINs universally and the emergence of mosquitoes able to survive exposure to pyrethroid insecticide – historically, the principle active ingredient for LLINs – has led malaria researchers to question the protection offered by insecticide [10, 11]. Others strongly advocate that the killing effect of LLINs is integral to their continual protective benefit, more so than universal coverage of a community [12]. LLINs offer different benefits for users and non-users within a community: net-users receive personal protection while non-users benefit from indirect protection. This is due to the reduced numbers of mosquitoes, and reduced proportion of these mosquitoes which are infectious (due to higher mosquito mortality and lower human infectious reservoir) [13–15]. This logic, shown empirically [16, 17] and theoretically using mathematical models [13, 15], formed the basis for the adoption of universal coverage with LLINs as a global policy by the World Health Organisation (WHO) [18, 19]. It is possible to quantify the direct and indirect protection offered from LLINs to both users and non-users further into benefits afforded by the barrier distinctly to benefits from the insecticide. Doing so can contribute to the debate on the use of untreated nets [10, 11] and can inform potential loss in impact due to pyrethroid resistance in local mosquito vectors.”

We update the discussion similarly on L312-322:

“The work adds to the existing evidence-base for the presence of a community effect. Many experimental hut trial have shown that mosquito feeding attempts are altered by the presence of insecticide treated nets in rooms [30, 38–40] and randomised control trials demonstrate that this altered behaviour – including increased mosquito mortality - is leading to significant public health benefits from nets [20–22]. Early trials indicated that non-users were also afforded protection from these actions on mosquito densities and behaviours [22, 41]. It was our aim here to try to disentangle and quantify the direct and indirect protection offered

from LLINs to both users and non-users into benefits afforded by the barrier distinctly to benefits from the insecticide. This allows us to estimate what is lost through the presence of pyrethroid resistant local mosquitoes that reduces the well-established critical role of the killing effect of insecticide-treated nets [12, 13, 15, 17].”

And L393-406:

“Results here and elsewhere [9, 19, 25, 61] illustrate the importance of achieving high coverage with both LLINs and untreated nets as increasing net use generally will be beneficial for community protection (Figure S7). Killeen [19] argues that prioritising vector mortality rather than universal usage may be more cost-effective if newer products, with alternative active ingredients, are more expensive (and therefore fewer can be procured for a given region). This makes sense if the killing effect is particularly potent, and nets are distributed in a way that means non-users are in close enough proximity to benefit from the reduced numbers of mosquitoes caused by the killing action of the net. Novel net designs that either chemically [62–64] or mechanically [65] kill mosquitoes are welcomed in order to mitigate for the potential loss in impact from pyrethroid-LLINs [31, 66]. It may be cost-effective to then top-up communities with untreated nets to provide some barrier protection to those members who do not have the treated net. A cost-effective analysis is beyond the scope of the current work, particularly because it becomes logistically harder and more expensive to increase usage once net use has reached around 60% so the association is non-linear [67], but certainly worth consideration for future discussion.”

And we have adjusted the abstract to highlight the novel aspects of the current submission on L11-35:

“Long lasting insecticidal nets (LLINs) provide both direct and indirect protection against malaria. LLINs, historically including pyrethroid active ingredients, are designed to provide protection at the community level, while also protecting individuals directly. As pyrethroid resistance evolves in mosquito vectors across malaria-endemic countries, it will be useful to understand how the specific benefits LLINs afford individuals and communities may be affected. We consider direct protection from the barrier and insecticide for the net-user, and indirect protection from the barrier and insecticide for non-users of nets. It is challenging and unethical to empirically test differences because it could involve placing individuals at unnecessary risk of exposure to infectious bites. Instead, we use a modelling framework and compare predictions to trends with Demographic Health Survey (DHS) data.

Our modelling exercise predicts that in a hypothetical situation with an entomological inoculation rate (EIR) of 100 infectious bites per person per year, the reduction in EIR from an untreated net used by 80% of the population is 42% [95% CI: 11% - 71%] for users and 12% [95% CI: -3% to 40%] for non-users. In this scenario, due to the impact of the insecticide, the reduction in EIR due to LLINs is 89% [95% CI: 67% - 98%] for users and 74% [95% CI: 48% - 92%] for non-users, but this protection reduces as insecticide resistance in mosquitoes increases. The DHS data indicates similar benefit for net-users to that simulated by the model, but we were unable to detect an impact from increasing pyrethroid resistance in these data.

We find that there is no minimum LLIN usage needed for users and non-users to benefit from community protection. Modelling results indicate that pyrethroid resistance in local

mosquitoes will likely diminish the direct and indirect benefits from insecticides, leaving the barrier effect for net-users intact, but LLINs are still expected to provide enhanced benefit over untreated nets even at high levels of pyrethroid resistance.”

3. Similarly, statement like the following on line 52 of the introduction are simply inaccurate and seem to ignore all the fine details of all the many papers published over the years: “Evidence of the community effect is ambiguous, partly because empirical studies have not been powered to address this question – it would be unethical now to deliberately withhold nets from a cohort of a community.” Indeed, it certainly would, but there are also many folks who simply decline to use them even when provided freely and many other examples where this information can be gleaned from field studies, most notably the spatially explicit analysis in Hawley et al 2003. Similarly, all the statements in the paragraph beginning on line 63 about community level effects being impossible to define, much less measure, are inaccurate and appear intended to over-represent the novelty of the work presented. Similarly, while the new analysis described in the final paragraph of the introduction is certainly valuable, the statement “which is not possible to do from trial data” goes too far.

We apologise for the lack of scrutiny we have given to previous work. We do think however that it is not possible to quantify the 5 categories we have isolated (barrier effect, direct benefit from insecticide for net users, indirect barrier effect and indirect benefit from insecticide for non-users of nets) using trials as these assess impacts – rightly so – at the population level.

We have rewritten the introduction, we hope, with greater understanding of previous work and a better explanation of the contribution we are making. Please see our answer above and also L84-113:

“Evidence is now building that pyrethroid resistance in mosquito vector populations is leading to diminished protection from LLINs that use this insecticide as the active ingredient [28–31]. In parallel, evidence is emerging that different net brands may be more robust than others, potentially offering longer direct benefits to net users than others [32]. Both the barrier and the insecticide of the net offer direct and indirect protection to both users and non-users. However, these may be differently impacted by the presence of pyrethroid resistance in local mosquito populations or the integrity of the netting material. In this context, it could be useful to quantify the different types of benefit offered by LLINs so that we can start to consider how to mitigate against lost personal or community protection by coupling nets falling short with alternative interventions or focusing research and development efforts on enhancing particular benefits [12]. It would be unethical to test these distinctions empirically because it would require leaving a cohort of people without nets and therefore exposed to potential transmission risks. In addition, it is difficult to isolate the barrier and insecticidal benefit offered to a single individual in a community. To overcome these problems, we chose to investigate this question by adapting an established mechanistic model that tracks *Plasmodium falciparum* malaria between mosquitoes and people. However, we also wanted to explore whether community level outcomes simulated where reasonably embedded in reality. Demographic Health Survey (DHS) data [33] provide a potential opportunity to explore differences in prevalence between individuals using nets and those choosing not to, or being unable to do so (e.g. due to access limitations), from within the same communities. These data are produced every few years from sentinel settings and summarise, among many other aspects of health: i) the proportion of people in

the assessed cluster using nets, ii) those having access to nets, and iii) the prevalence of malaria parasite infections detected by rapid diagnostic test at the individual level.

In this manuscript, we compared the difference in prevalence between users and non-users of LLINs in a mechanistic transmission model of falciparum malaria [34–37]. To provide some confidence in the model we statistically analyse DHS data to explore whether qualitative predictions made by the model are supported by epidemiological evidence. We then use the transmission model to tease apart the direct benefit of LLINs from the mass community effect and investigate what happens as mosquitoes show increasing resistance to pyrethroid insecticide. We discuss these findings in the context of previous work quantifying personal protection and the community effect.”

4. The actual analysis presented in the results section is quite neat and impressive and does add some nice depth and nuance to the evidence base, so this remains a useful paper worthy of publication, but the abstract and introduction need to be reframed around what this study does add rather than exaggerate the novelty of simply distinguishing community and personal protection effects.

We hope our restructured abstract and introduction better place the work into context.

5. Lines 56 onwards: The apparent lack of an obvious community effect in The Gambia is most probably the result of very high mosquito dispersal ranges in that setting, which cause effects to be diluted out through mixing of mosquito populations across treatment arm villages, as demonstrated in Hawley et al 2003 among others and discussed in Killeen et al 2003 Lancet Infect Dis 3: 297. Again, looks intended to create the impression of a long unsolved mystery that is misrepresentative and exaggerates the novelty and usefulness of this piece.

Apologies – please see the rewritten introduction that better places the work in context of previous contributions.

6. Similarly, in the discussion, outline the role of the insecticide in delivering on the impact of LLINs is not as novel as portrayed and these issues, as well as the effect of resistance and potential impact of resistance management options have been discussed at length in other pieces in authoritative journals in recent years (Briet et al 2013 Malar J 12: 77, Viana et al 2016 PNAS 113: 8975, Killeen 2020 and refs therein). In particular, much bigger picture strategic issues over the short and long term have been explored in a more fundamental way in Killeen 2020). While I certainly agree with the importance of countering the recently initiated and very silly “conversation on the necessity for insecticide in nets” mentioned on line 287, and welcome this contribution to the evidence base that does so, this paper would be much stronger if it presented itself as such, rather than an entirely unprecedented collection of new insights.

Please see the rewritten discussion where we have better contextualised the contribution with previous knowledge.

MINOR COMMENTS

7. In the opening of the abstract and introduction, the full name for LLINs should exactly match the acronym. If necessary, say “Long-lasting insecticide-treated mosquito nets, or more simply, long lasting insecticidal nets (LLINs)...”.

Thanks, we have changed in both places to be “Long lasting insecticidal nets”.

8. Line 44: “never-the-less” should be a single, unbroken word.
Thanks, fixed.

REVIEWER COMMENTS

Reviewer #1 (Remarks to the Author):

The revised version of the manuscript is substantially improved and all comments have been sensibly and adequately addressed.

Reviewer #2 (Remarks to the Author):

I'm still struggling with the lack of quantitative reconciliation between the DHS analysis and the modeling predictions. Figure 1C and 1D suggests that the model overpredicts the gap between users and non-users and thus I'm not sure what to make of the modeling results at all: are they meant to be qualitative? Yet the main claims of the paper, as listed in the abstract, are quantitative.

Response to R2 #14: Quite confused by this response. Yes, taking an estimate from the literature and applying it to the DHS clusters won't match the exact setting for each cluster. But assuming perfect usage also doesn't match the setting for each cluster. Since the model is overpredicting the gap in prevalence between users and non-users, why not test how relaxing the assumption of perfect usage, using approximate data from research studies, might improve the quantitative agreement between the model and the DHS analysis?

Response to R2 #15: Sure, a hypothetical scenario wouldn't have a clear empirical parallel, but empirical data should be able to support or challenge the claim that 74-89% EIR reduction from $EIR = 100$ is a plausible outcome of LLIN distribution at 80% use.

Why is net retention modeled with a half-life of 5 years with Bertozzi-Villa et al 2021 suggests that it's typically much shorter, perhaps 1.5-2 years?

What does this mean: "We assume a LLIN user always uses their net, with a three-year waning cycle." Is the 3-year waning cycle different from the 5-year retention half-life? What is waning here if not usage?

Reviewer #3 (Remarks to the Author):

AUTHOR REBUTTAL COMMENT: "It is our understanding that it is difficult to empirically quantify a community effect. Were we to try to estimate this through gold standard experimental hut, randomised control trials, or other experimental frameworks, we would be unable to isolate the barrier effect of the net for the user i.e. the degree that the barrier acts to isolate the net-user from local mosquito bites."

REVIEWER RESPONSE: Look in the details of all the old community-scale RCTs trials of ITNs, specifically in the negative controls: the incidence and prevalence rates among the minority of net users in those clusters despite lack of promotion, subsidization or provision give direct estimates of the barrier effect, with the community-level effects being the difference between these subjects and those living in the intervention clusters. I must say this comment confirms my impression of a failure to pay attention to old literature, in particular the details of the primary reports.

AUTHOR REBUTTAL COMMENT: "We can only measure population level effects on various mosquito outcomes that inform transmission metrics like the vectorial capacity. Even with the Hawley et al. (2003) publication, which is an excellent early study supporting the presence of a community effect from ITNs, we cannot isolate the different protections offered to individuals across the population. Our attempt here is to do this so that we can try to understand the different effects that ITNs can have, particularly to try to quantify the barrier effect and then the indirect benefits from the induced killing of mosquitoes that is of course a key action of insecticidal nets."

REVIEWER RESPONSE: This comment further consolidates my impression that old evidence is not only being neglected, it is being ignored and misrepresented. For example figure 2 in Hawley et al

demonstrates and provides numerical estimates for the impact of purely community-level effects up to 300m away from the boundary with ITN intervention clusters and table 3 of the same paper puts hard numbers on barrier effects, summarized as follows in the title of the table:

“Comparison of estimates of insecticide-treated bed net (ITN) efficacy before and after adjustment for presence of a community effect in control compounds within 300 meters of intervention compounds”

AUTHOR REBUTTAL COMMENT:

“The Killeen et al. 2007 publication presents a method to identify the individual protection offered by a mosquito net and the protection.....etc etc.....In our approach, we consider the barrier effect and the indirect benefit from the insecticide to a net-user, but then separate the community benefit further into the direct and indirect benefits to others not using nets due to the insecticide and collate these as an overall benefit. In our opinion, this is useful particularly in the context of increasing pyrethroid resistance across malaria-endemic countries because we can use the process to quantify how these benefits may be affected.....etc etc.”

REVIEWER COMMENT: Again, these subtleties are useful but largely far from unprecedented and objective consideration of the literature would reflect that. For example, further developments of the Killeen et al models cover most of these functionality issues (Killeen et al 2011 Target product profile choices for intradomestic malaria vector control pesticide products: repel or kill? Malaria Journal 10:207) and then apply them specifically to the resistance issue in the Killeen 2020 reference in my previous review. Other examples include Briët et al. 2013 (Effects of pyrethroid resistance on the cost effectiveness of a mass distribution of long lasting insecticidal nets: a modelling study. Malaria Journal. 12:77. I would have strongly preferred a substantive and objective manuscript revision to this very long-winded and evasive rebuttal.

AUTHOR REBUTTAL COMMENT: “We apologise for the lack of scrutiny we have given to previous work. We do think however that it is not possible to quantify the 5 categories we have isolated (barrier effect, direct benefit from insecticide for net users, indirect barrier effect and indirect benefit from insecticide for non-users of nets) using trials as these assess impacts – rightly so – at the population level.

... etc etc

... It would be unethical to test these distinctions empirically because it would require leaving a cohort of people without nets and therefore exposed to potential transmission risks.”

REVIEWER COMMENT: See my comments on the poor consideration of old literature above- indeed it would be unethical today but fortunately this evidence was collected in the old RCTs conducted before universal scale up of ITNs and the LLINs.

AUTHOR REBUTTAL COMMENT: “We hope our restructured abstract and introduction better place the work into context.”

REVIEWER COMMENT: Indeed the revised abstract is a more reasonable and objective representation of the work. This paper does present new tools and new evidence but their novelty and impact on our understanding shouldn't and doesn't need to be exaggerated.

AUTHOR REBUTTAL COMMENT: Apologies – please see the rewritten introduction that better places the work in context of previous contributions.”

REVIEWER COMMENT: Indeed, this is a welcome improvement but still underplays what has been known for many years on this topic. Of course, the Killeen et al models are only models but the empirical evidence base generated specifically to support the transition to universal coverage policies are 100% consistent with these mechanistic insights. Please see above my comments about neglect and misrepresentation of the old empirical literature.

ADDITIONAL REVIEW COMMENT: References 16 and 21 are duplicates of the same paper, as are references 12 and 19 of another. Looks unacceptably rushed and lacking in adequate editing checks.

Response to reviewers

Reviewer comments are shown in black. Our responses are indicated in blue.

Reviewer #1 (Remarks to the Author):

The revised version of the manuscript is substantially improved and all comments have been sensibly and adequately addressed.

Thank you.

Reviewer #2 (Remarks to the Author):

I'm still struggling with the lack of quantitative reconciliation between the DHS analysis and the modeling predictions. Figure 1C and 1D suggests that the model overpredicts the gap between users and non-users and thus I'm not sure what to make of the modeling results at all: are they meant to be qualitative? Yet the main claims of the paper, as listed in the abstract, are quantitative.

We thank you for this comment and agree as previously written it was confusing how the results should be interpreted. We believe some of the discrepancies between the DHS data and model predictions are due to not being able to capture heterogeneities within the clusters, such as house type, that are independent of net usage. We have added a sentence to reflect this in the discussion on L338-342 of the clean version:

"Taking the lowest possible spatial scale (clusters), we can see that the broad qualitative conclusions identified by the mechanistic model are statistically supported by the observed data. However, we were unable to capture the heterogeneities within clusters that are inherent in natural field settings and likely to be expressed within the DHS data."

We have also reduced the precision of the numbers we report in our abstract to reflect the nature of using a hypothetical scenario. The abstract on L22-29 now says:

"Our modelling exercise predicts that in a hypothetical situation with an entomological inoculation rate (EIR) of 100 infectious bites per person per year, the reduction in EIR from an untreated net used by 80% of the population is around 40% for users and around 10% for non-users. However, estimates are highly uncertain. In this scenario, due to the impact of the insecticide, the reduction in EIR due to LLINs is around 90% for users and around 70% for non-users, but this protection reduces as insecticide resistance in mosquitoes increases. Similar trends are obtained for other settings. The DHS data indicates similar benefit for net-users to that simulated by the model, but we were unable to detect an impact from increasing pyrethroid resistance in these data."

Finally, we have gone into detail why the qualitative predictions might match but the scale of the magnitude might not on L385-392.

"As our model assumes systematic net usage the difference between ITN users and non-users it predicts should probably be seen as the maximum that is likely to occur, with heterogenous usage and transmission likely to diminish this difference in practice. So, though the model is able to capture the broad qualitative conclusions, the absolute magnitude of these differences should be treated with caution. Nevertheless, whether the model captures these differences should also not substantially influence our quantification of the direct and indirect benefits of LLINs, which rely on similar structural assumptions, just our ability to match the model results to field data."

Response to R2 #14: Quite confused by this response. Yes, taking an estimate from the literature and applying it to the DHS clusters won't match the exact setting for each cluster. But assuming perfect usage also doesn't match the setting for each cluster. Since the model is overpredicting the gap in prevalence between users and non-users, why not test how relaxing the assumption of perfect usage, using approximate data from research studies, might improve the quantitative agreement between the model and the DHS analysis?

Thank you for your comment and sorry for the confusion. Since we use a compartmental model for the main analysis in our paper splitting out the different component of protection offered by bed-nets, it is not possible to relax the assumption of perfect usage by changing an individual's net usage category throughout the simulation. People are either in the user compartment or non-user compartment. Relaxing the assumption would require an individual based model that would not enable us to do the rest of the analysis. We agree with you that if we were to repeat this analysis with relaxing the assumption of perfect usage, we may get better agreement between the model and data, but we could not achieve the second part of our analysis. We adapted our sentences on this on L381- 384 in the discussion:

“Parameterising an individual-based model with field estimates of this inter and intra season, nightly, and within-cluster variability could substantially reduce the discrepancy between observed data and model predictions and could enable us to capture within cluster heterogeneities.”

Response to R2 #15: Sure, a hypothetical scenario wouldn't have a clear empirical parallel, but empirical data should be able to support or challenge the claim that 74-89% EIR reduction from EIR = 100 is a plausible outcome of LLIN distribution at 80% use.

We thank you for further clarifying your original comment here. We found a review paper¹ looking at the reduction in EIR from different vector control scenarios. Despite this paper finding 6 studies which used insecticide treated nets, none of them mention the all-age LLIN usage achieved during the study period. For these studies percentage reductions in EIR were between -42 to 97% with all but one between 55 and 97%. This range encompasses our findings. In Lindblade et al (2004)², there was a 91% reduction in EIR for a 65.9% usage in children under 5. The control villages had a EIR of 0.93 bites per person per month. This suggests high percentage reductions are possible with the use of bed-nets although it's difficult to know what all-age usage for this community would be. It's likely that this result is for a lower all-age usage than 80%.

We have added a few additional sentences to the discussion on L326-330:

“A review by Shaukat et al. [41] identified six studies that involved insecticide treated nets, but information about all-age usage levels were incomplete. For these studies percentage reductions in EIR were between -42% to 97% with all but one between 55 and 97%. Lindblade et al. [42] found there was a 91% reduction in EIR for a 65.9% usage in children under 5.”

Why is net retention modeled with a half-life of 5 years with Bertozzi-Villa et al 2021 suggests that it's typically much shorter, perhaps 1.5-2 years?

We do not model net retention, which is a limitation of our model, and assume that everyone who is given a net uses it and just the efficacy of it decays with a half life $\gamma_N = 2.640$ years. We have added this as a limitation on L367-369:

“A limitation of our model is that it assumes systematic use of LLINs over time, i.e., users will always use a LLIN each night whilst non-users will not, though community use is simulated to wane with time since the previous mass campaign.”

What does this mean: “We assume a LLIN user always uses their net, with a three-year waning cycle.” Is the 3-year waning cycle different from the 5-year retention half-life? What is waning here if not usage?

We are sorry this is not clear. We assume the performance of the net decays with $\gamma_N = 2.640$ years but that nets are replaced every 3 years. We have changed the text on L522 to say:

“We assume a LLIN user always uses their net, with a three-year replacement cycle.”

Reviewer #3 (Remarks to the Author):

AUTHOR REBUTTAL COMMENT: “It is our understanding that it is difficult to empirically quantify a community effect. Were we to try to estimate this through gold standard experimental hut, randomised control trials, or other experimental frameworks, we would be unable to isolate the barrier effect of the net for the user i.e. the degree that the barrier acts to isolate the net-user from local mosquito bites.”

REVIEWER RESPONSE: Look in the details of all the old community-scale RCTs trials of ITNs, specifically in the negative controls: the incidence and prevalence rates among the minority of net users in those clusters despite lack of promotion, subsidization or provision give direct estimates of the barrier effect, with the community-level effects being the difference between these subjects and those living in the intervention clusters. I must say this comment confirms my impression of a failure to pay attention to old literature, in particular the details of the primary reports.

¹ <https://malariajournal.biomedcentral.com/track/pdf/10.1186/1475-2875-9-122.pdf>

² <https://europepmc.org/article/MED/15173148>.

We are sorry for not being clear with the above rebuttal comment and not highlighting carefully enough the early RCT literature. We had not intended to say there was no evidence for the different components of ITN protection, just that measuring it is difficult/costly and has only been conducted a few times, making it hard to compare the importance of other covariates such as net use and pyrethroid resistance. We have amended our introduction on L72-88 to include more detail:

"Most studies measure the overall efficacy of LLINs through cluster randomised control trials [13, 20–22]. We consider direct protection to the net-user from two components: direct protection attributed to the barrier and direct protection attributed to the effects from the insecticide that kills or deters mosquitoes from biting the protected individual. The combination of both these terms was referred to as the barrier effect in some of the previous literature, but here we separate it for the actions of the insecticide (i.e. the barrier effect but not the insecticidal actions are seen in untreated nets). The whole community benefits indirectly from the barrier and the insecticide given lower burden of infection across the community. Previously the contribution from both these terms has been referred to as the community effect. These different levels of benefit are difficult to measure empirically though the evidence has recently been summarised [23]. One early study providing evidence of the community effect comes from a cluster randomised trial of insecticide treated nets (ITN) conducted in western Kenya [13]. In this trial a clear gradient of impact was observed in the control areas in which ITNs were not distributed but that were close neighbours of areas in which ITNs were distributed, with a reduction across several different malaria-related outcomes including malaria prevalence and parasitaemia. ITN usage in the intervention areas was observed to be around 70% [24]. Other early trials enable the estimation of different components of protection e.g. direct protection from barrier + insecticide was quantified in [13, 34, 35], direct protection from the barrier in [36], and direct protection from insecticide in [37] (Table 1)."

In citations 13 (Hawley et al.), 34 (Lindblade et al.) and 35 (Beach et al.) the malaria outcomes are compared between ITN users and non-users.

In citation 36 (Clarke et al.) malaria outcomes are compared between children using untreated nets and no nets. In citation 37 (Snow et al.) malaria outcomes are compared between children using treated nets and untreated nets.

In citation 13 (Hawley et al) community effect sizes are also given.

We have also added the following to the discussion on L325-330

"Early trials indicated that non-users were also afforded protection from these actions on mosquito densities and behaviours [13, 46, 47]. A review by Shaukat et al. [48] identified six studies that involved insecticide treated nets, but information about all-age usage levels were incomplete. For these studies percentage reductions in EIR were between -42% to 97% with all but one between 55 and 97%. Lindblade et al. [34] found there was a 91% reduction in EIR for a 65.9% usage in children under 5."

AUTHOR REBUTTAL COMMENT: "We can only measure population level effects on various mosquito outcomes that inform transmission metrics like the vectorial capacity. Even with the Hawley et al. (2003) publication, which is an excellent early study supporting the presence of a community effect from ITNs, we cannot isolate the different protections offered to individuals across the population. Our attempt here is to do this so that we can try to understand the different effects that ITNs can have, particularly to try to quantify the barrier effect and then the indirect benefits from the induced killing of mosquitoes that is of course a key action of insecticidal nets."

REVIEWER RESPONSE: This comment further consolidates my impression that old evidence is not only being neglected, it is being ignored and misrepresented. For example figure 2 in Hawley et al demonstrates and provides numerical estimates for the impact of purely community-level effects up to 300m away from the boundary with ITN intervention clusters and table 3 of the same paper puts hard numbers on barrier effects, summarized as follows in the title of the table:

"Comparison of estimates of insecticide-treated bed net (ITN) efficacy before and after adjustment for presence of a community effect in control compounds within 300 meters of intervention compounds"

We thank you for your comment and are sorry that our definitions of what we are trying to calculate were not clear. We have made our definitions of the barrier effect clearer on L73-79:

"However, we consider direct protection to the net-user from two components: direct protection attributed to the barrier and direct protection attributed to the effects from the insecticide that kills or deters mosquitoes from biting the protected individual. Here we separate the barrier from the actions of the insecticide (i.e. the barrier effect but not the insecticidal actions are seen in untreated nets). The whole community benefits indirectly from the barrier and the insecticide given lower burden of infection across the community. Previously the contribution from both these terms has been referred to as the community effect."

We agree Hawley clearly demonstrates the combination of the indirect barrier effect and indirect insecticide for non-net users, but it cannot attribute what proportion community effect comes from the insecticide on the bed nets and the barriers themselves.

AUTHOR REBUTTAL COMMENT:

“The Killeen et al. 2007 publication presents a method to identify the individual protection offered by a mosquito net and the protection.....etc etc.....In our approach, we consider the barrier effect and the indirect benefit from the insecticide to a net-user, but then separate the community benefit further into the direct and indirect benefits to others not using nets due to the insecticide and collate these as an overall benefit. In our opinion, this is useful particularly in the context of increasing pyrethroid resistance across malaria-endemic countries because we can use the process to quantify how these benefits may be affected.....etc etc.”

REVIEWER COMMENT: Again, these subtleties are useful but largely far from unprecedented and objective consideration of the literature would reflect that. For example, further developments of the Killeen et al models cover most of these functionality issues (Killeen et al 2011 Target product profile choices for intradomestic malaria vector control pesticide products: repel or kill? Malaria Journal 10:207) and then apply them specifically to the resistance issue in the Killeen 2020 reference in my previous review. Other examples include Briët et al. 2013 (Effects of pyrethroid resistance on the cost effectiveness of a mass distribution of long lasting insecticidal nets: a modelling study. Malaria Journal. 12:77. I would have strong preferred a substantive and objective manuscript revision to this very long-winded and evasive rebuttal.

We are sorry with our inconsistency in our definitions of the direct and indirect effect that has led to a very confusing rebuttal above. This sentence "*In our approach, we consider the barrier effect and the indirect benefit from the insecticide to a net-user, but then separate the community benefit further into the direct and indirect benefits to others not using nets due to the insecticide and collate these as an overall benefit*" can more clearly be written as:

“In our approach, we separate the community benefit into 4 components: the direct impact of the barrier, the direct impact of the insecticide, the indirect impact of the barrier and the indirect impact of the insecticide (where the first two only impact net users)”

We have further clarified the definitions by the augmentation of Table 1.

Table 1: Definition of the four different mechanisms determining the overall level of ITN protection and the equations used for their estimation. The four mechanisms (defined A-D) are described according to their impact on transmission. Direct benefits occur to those using ITNs, whereas indirect benefits act upon everyone in a community where ITNs are in use. The value of each are estimated from a transmission dynamics mathematical model, with the different scenarios (*italic lower case letters*) refer to model run with different assumptions presented in Figure 2.

	Direct Net users	Indirect (community effect) Net users and non-users
Barrier	(A) Reduction in EIR due to reduced mosquito blood-feeding rate. Calculate as control – a	(B) Reduction in EIR for net users and non-users caused by lower human prevalence resulting from (A). Calculated as a – b
Additional protection from insecticide	(C) Further reduction in EIR caused by insecticide killing and deterrence of blood-seeking mosquitoes Calculated as a – c	(D) Further reduction in EIR due to higher mosquito mortality, lower mosquito abundance and lower human prevalence resulting from (C) Calculated as b – d – (a – c)

We have clarified our definitions as stated in the previous comment and checked our language throughout the manuscript for clarity. We do cite Killeen et al 2011, Killeen 2020 and Briët et al as follows on L60-70:

“As shown in previous research such as Hawley et al. [13] and Killeen et al. [14, 15] LLINs offer different benefits for users and non-users within a community: net-users receive personal protection while non-users benefit from indirect protection. This is due to the reduced numbers of mosquitoes, and reduced proportion of these mosquitoes which are infectious (due to higher mosquito mortality and lower human infectious reservoir) [15–17]. This logic, shown empirically [13, 18] and theoretically using mathematical models [15, 17], formed the basis for the adoption of universal coverage with LLINs as a global policy by

the World Health Organisation (WHO) [12, 19]. It is possible to quantify the direct and indirect protection offered from LLINs to both users and non-users further into benefits afforded by the barrier distinctly to benefits from the insecticide. Doing so can contribute to the debate on the use of untreated nets [10, 11] and can inform potential loss in impact due to pyrethroid resistance in local mosquito vectors.”

AUTHOR REBUTTAL COMMENT: “We apologise for the lack of scrutiny we have given to previous work. We do think however that it is not possible to quantify the 5 categories we have isolated (barrier effect, direct benefit from insecticide for net users, indirect barrier effect and indirect benefit from insecticide for non-users of nets) using trials as these assess impacts – rightly so – at the population level.

... etc etc

... It would be unethical to test these distinctions empirically because it would require leaving a cohort of people without nets and therefore exposed to potential transmission risks.”

REVIEWER COMMENT: See my comments on the poor consideration of old literature above- indeed it would be unethical today but fortunately this evidence was collected in the old RCTs conducted before universal scale up of ITNs and the LLINs.

We are sorry that you did not feel our summary of previous RCT trials were strong enough. We have no added more information about the Hawley trial on L80-86:

“One early study providing evidence of the community effect comes from a cluster randomised trial of insecticide treated nets (ITN) conducted in western Kenya [13]. In this trial a clear gradient of impact was observed in the control areas in which ITNs were not distributed but that were close neighbours of areas in which ITNs were distributed, with a reduction across several different malaria-related outcomes including malaria prevalence and parasitaemia. ITN usage in the intervention areas was observed to be around 70%”

AUTHOR REBUTTAL COMMENT: “We hope our restructured abstract and introduction better place the work into context.”

REVIEWER COMMENT: Indeed the revised abstract is a more reasonable and objective representation of the work. This paper does present new tools and new evidence but their novelty and impact on our understanding shouldn't and doesn't need to be exaggerated.

Thank you.

AUTHOR REBUTTAL COMMENT: Apologies – please see the rewritten introduction that better places the work in context of previous contributions.”

REVIEWER COMMENT: Indeed, this is a welcome improvement but still underplays what has been known for many years on this topic. Of course, the Killeen et al models are only models but the empirical evidence base generated specifically to support the transition to universal coverage policies are 100% consistent with these mechanistic insights. Please see above my comments about neglect and misrepresentation of the old empirical literature.

We are sorry you feel we have neglected and misrepresented the empirical literature. We have edited our rewritten introduction in line with your comments as explained above to make the previous contributions stronger.

ADDITIONAL REVIEW COMMENT: References 16 and 21 are duplicates of the same paper, as are references 12 and 19 of another. Looks unacceptably rushed and lacking in adequate editing checks.

We are sorry about the duplicated references that occurred due to multiple people working on the manuscript. We have fixed these and ensured no more references are duplicated.

REVIEWERS' COMMENTS

Reviewer #2 (Remarks to the Author):

Major comments:

Main results (point estimates) are with assumption of 85% probability of a mosquito bite in bed in the absence of nets: this is not mentioned anywhere in the main text and needs to be dug out from the supplement. Since the modeled impact on EIR is so high, this assumption should be highlighted throughout, especially since behavioral resistance, outdoor biting, people going to bed later, etc. are all important points of discussion right now around LLINs.

Results section in general: this aspect of your finding that increasing coverage of the barrier results in reduced impact of the net on users is a little confusing and I wonder if there's a way to make it clearer to the reader. I think what is happening is that you are looking at 3 or so vector-related quantities: # bite attempts, # successful bites, and # successful infectious bites. As the fraction of people protected by a barrier increases, # attempts also increases, and there is propagation down the chain. It is confusing though in scenario a why the non-users would not also be a sink for the extra bite attempts. Maybe they are, but EIR is artificially maintained constant? This is not clear. See also below on other places where the attempts and propagation are confusing.

For the Discussion, why is there no contextualization of the model predictions of indirect effects with control arm or spillover effects from net trials (as per Reviewer #3's comments)? I see some contextualization with intervention arm results, which is great, but no numbers are mentioned for empirical observations of plausible indirect effects.

On claims of quantitative effect size and lack of quantitative agreement with field data: I agree that certain aspects of the DHS data (definition of user in DHS analysis vs in model, for example) could definitely be driving some, or all, of the lack of quantitative agreement. But I am really confused by these sentences: "As our model assumes systematic net usage the difference between ITN users and non-users it predicts should probably be seen as the maximum that is likely to occur, with heterogenous usage and transmission likely to diminish this difference in practice. So, though the model is able to capture the broad qualitative conclusions, the absolute magnitude of these differences should be treated with caution. Nevertheless, whether the model captures these differences should also not substantially influence our quantification of the direct and indirect benefits of LLINs, which rely on similar structural assumptions, just our ability to match the model results to field data." What am I to make of this? The absolute magnitude is stated elsewhere in the paper without caveats on assumptions on vector behavior and net retention (I don't understand why more realistic net retention could not be implemented in the model, it seems fairly straightforward to me). The ITN effect size overall seems much rosier in this model than in others, for example the model shown in Fig 9.9 of the 2022 World Malaria Report has effects on prevalence nearly gone by the end of 3 years for a realistic net campaign.

Since indirect effects of nets have already been approximated both with models and with control/spillover trial arms, I'm not sure what new knowledge this paper adds. The authors could have used an individual-based model and tried to understand what role heterogeneity and inconsistent usage might play in modulating the effects of LLINs, which are otherwise predicted to be very strong, perhaps much stronger than have been observed in programmatic implementation. That kind of model could be really helpful for programs and the global community to understand why nets have not had more of an impact. What is the second part of the analysis that is not achievable with an individual-based model (from authors' rebuttal)?

Minor comments:

L61-2: don't net users also receive indirect protection?

L146-7: please put the peak difference here, otherwise it seems like you're trying to hide it

Fig 1C: please add to the caption that the circle fill color is the difference in DHS. At first glance it's hard to see that there is any color at all and I was confused by L152-3 saying that Fig 1C shows cluster-level data. Please also clarify that net usage is all-age net usage, not U5 net usage (I think

this is the case from reading the Methods)

L177-8: what is meant by "magnitude of the benefit provided by the barrier and the community effect"? from your results, the barrier can have a community effect, so this phrasing is confusing

L179-82: I found these sentences confusing, in addition to a control there are 4 scenarios and a control (so a total of 1+4+1=6 scenarios)? But from Fig 2 it looks like there are a total of 5 scenarios?

L213-4: does the 11% EIR reduction to the community include non-users, whose EIR is fixed? Is it that their EIR would go down (by some amount not specified) if allowed to decrease? This is confusing

L244-8: a little confusing to me how this is not rolling in some indirect benefit to users.

L249: how is it the "same pattern"? it looks like the opposite pattern to me (more vs less benefit at high coverage)?

L252-3: isn't the effect on attempts also included in scenario a? how can there be even more attempts when there is insecticide? Is it due to the repellency effects?

L279-82: I thought there was zero benefit to non-users when only 1 person uses a net? Perhaps rephrase this sentence (?)

L315-6: greater relative reductions in EIR seen at higher levels of resistance: this is the opposite of what Fig 4 shows, no?

L350-3: sentence beginning "Similarly" is confusing. Very little difference in prevalence between users and non-users? A change in prevalence in each of the two groups, or a change in the difference in prevalence between the two groups?

L359-65: for some of these factors, haven't you already shown that you think some have little effect?

L372: "routine data" I would not at all consider DHS data to be routine data. Routine data usually refers to information that comes from surveillance systems (cases, interventions, etc), and DHS is survey data not from routine systems.

L377-8: does "seasonality" refer to seasonality of net use, or seasonality of transmission?

L384-5: why is a broad geographical and temporal period needed for a validation dataset? Why not a research dataset that includes a high level of heterogeneity in the research site?

L380-1: are you saying exposure fluctuates annually like net use does (seasonally with vector abundance)? Or that there is variability in exposure across individuals living in the same area?

L417-8: are there any analyses of DHS data in the literature looking at net access within households and spatial variation in access that could inform on this?

L435-7: the DHS data are what they are, are you saying that your model result looking at the DHS data and pyrethroid resistance are especially uncertain? I think you are not interpreting the DHS data directly here, correct? Are your estimates of community resistance particularly "variable" or just very uncertain due to the nature of the resistance data and spatial scales you're working with?

Reviewer #3 (Remarks to the Author):

None

Response to reviewers

Reviewers' comments are written in black. Replies are indicated in blue

Reviewer #2 (Remarks to the Author):

Major comments:

Main results (point estimates) are with assumption of 85% probability of a mosquito bite in bed in the absence of nets: this is not mentioned anywhere in the main text and needs to be dug out from the supplement. Since the modeled impact on EIR is so high, this assumption should be highlighted throughout, especially since behavioral resistance, outdoor biting, people going to bed later, etc. are all important points of discussion right now around LLINs.

Thanks for this comment. We agree that this was an omission and have added this assumption to the methods section. L512-515 now reads:

“We assume a LLIN user always uses their net, with a three-year replacement cycle and that in the absence of control interventions 85% of mosquito bites are taken on people when they are in bed (estimates with LLINs will be higher and depend on net type, usage and time since deployment).”

Results section in general: this aspect of your finding that increasing coverage of the barrier results in reduced impact of the net on users is a little confusing and I wonder if there's a way to make it clearer to the reader. I think what is happening is that you are looking at 3 or so vector-related quantities: # bite attempts, # successful bites, and # successful infectious bites. As the fraction of people protected by a barrier increases, # attempts also increases, and there is propagation down the chain. It is confusing though in scenario a why the non-users would not also be a sink for the extra bite attempts. Maybe they are, but EIR is artificially maintained constant? This is not clear. See also below on other places where the attempts and propagation are confusing.

We agree that there is an awful lot going on here and that some of the results do feel counter intuitive. The reviewer is correct in that as more people have untreated nets, the benefit to users goes down because they start receiving mosquito bites repelled from other people also using a net. The confusion arises because non-users are also receiving these, but these are classed as indirect benefits of the barrier effect. In general, we have rewritten this section of the results, adding greater explanation, which we hope will aid interpretation. The main edited pieces are indicated in red.

“When one user in the population is protected, the direct impact from the barrier to that one user is a relative reduction in EIR of 29% [95% uncertainty interval (UI): 13% - 53%] (Figure 3a) from the original EIR of 100 infectious bites per person per year. Under the assumptions of the model, we express our findings as reductions in EIR relative to the starting EIR of 100 infectious bites per person per year so here, the effect size for any individual user would be a 29% relative reduction in infectious bites per year.

The level of EIR reduction from the barrier to a net-user does vary with usage level. As the usage increases, the direct protection from the barrier to those using an untreated net progressively reduces, when as many as 80% of the population are using nets, the direct protection for an individual from the barrier is a relative reduction in EIR of 14% [95% UI: 6% – 31%] (Figure 3a). This is due to the repellence action of the untreated barrier, which causes

the number of mosquito bites on users to increase as more people use nets. When there is only one net user in the population, they will be highly unlikely to receive any bites from mosquitoes which have attempted to feed elsewhere. As usage in the population increases, the overall number of mosquito feeding attempts will also increase, as those mosquitoes that previously tried to bite net users but were repelled search for a blood-meal elsewhere. These biting attempts are evenly distributed in the population so fall on both net users and non-users. As protection from untreated nets is partial, the EIR on users will therefore increase, reducing the direct benefit, even though the model assumes that some mosquitoes die (at their natural death rate) during the time it takes for a mosquito repelled by a net to searching for an alternative blood-meal (i.e., the probability of feeding and surviving decreases each feeding attempt as time passes). Non-users will also see an increase in the number of feeding attempts, though due to our definition of what constitutes a direct benefit these potential increases in biting are incorporated in the indirect section of Figure 3.

We can also summarise the direct benefit of the barrier to the combined community (users and non-users). When one person in the community uses a net, the relative reduction in EIR at the community level is minimal (0% [95% UI: 0% - 1%]). However, when 80% of the community are using nets, there is a relative reduction in EIR of 11% [95% UI: 5% - 25%] to the community.

Model predicted indirect benefit of barrier

The indirect benefit of the barrier is the protection provided to the community through lower malaria transmission due to people using untreated nets. In Figure 3 this protection is shown for users, non-users and the community in general according to the level of net usage. Using the model, we predict that indirect benefits from only one net in the community (scenario (A) – (B)) are minimal but that the magnitude of this type of protection increases with usage. When 80% of the community are using nets, we estimate a relative reduction in EIR of 24% [95% UI: 5% - 41%] for users, 12% [95% UI: -3% - 40%] for non-users and 22% [95% UI: 4% - 41%] for the community (Figure 3b). The impact of increasing untreated net usage on the indirect benefit to non-users is a trade-off between potentially receiving more mosquito bites (caused by mosquitoes being repelled from net users) vs the reduction in the probability that those bites are infectious (caused by fewer people in the community having malaria and the extended foraging time required to successfully blood feed). Our model suggests that for most scenarios, as usage increases, the indirect protection provided by the barrier effect increases (i.e. the reduction in the sporozoite rate has a greater impact on the EIR of non-users than the increase in their biting rate). However, there are some rare scenarios explored in the sensitivity analysis when the indirect benefit to non-users may be negative (as seen by some negative values in Figure 3b). Greater indirect benefits are seen in users rather than non-users because, as net usage increases, net-users experience less of an increase in mosquito bites caused by mosquitoes being dissuaded from feeding on users (conversely, non-users will receive more bites as net use increases as more mosquitoes will be repelled from users)."

For the Discussion, why is there no contextualization of the model predictions of indirect effects with control arm or spillover effects from net trials (as per Reviewer #3's comments)? I see some

contextualization with intervention arm results, which is great, but no numbers are mentioned for empirical observations of plausible indirect effects.

As stated in the paper, finding results which state the net usage and use similar measures of performance are hard to find in historical trial data. We are sorry if we have missed any papers you are aware of. We mention Hawley et al, which is a trial that shows evidence of indirect effects. We have added an initial sentence on L310-312:

“Hawley et al.[13] saw a similar odds ratio for clinical malaria and other malaria indicators in control compounds without bednets less than 300m from a border of compounds with nets.”

On claims of quantitative effect size and lack of quantitative agreement with field data: I agree that certain aspects of the DHS data (definition of user in DHS analysis vs in model, for example) could definitely be driving some, or all, of the lack of quantitative agreement. But I am really confused by these sentences: “As our model assumes systematic net usage the difference between ITN users and non-users it predicts should probably be seen as the maximum that is likely to occur, with heterogenous usage and transmission likely to diminish this difference in practice. So, though the model is able to capture the broad qualitative conclusions, the absolute magnitude of these differences should be treated with caution. Nevertheless, whether the model captures these differences should also not substantially influence our quantification of the direct and indirect benefits of LLINs, which rely on similar structural assumptions, just our ability to match the model results to field data.” What am I to make of this? The absolute magnitude is stated elsewhere in the paper without caveats on assumptions on vector behavior and net retention (I don’t understand why more realistic net retention could not be implemented in the model, it seems fairly straightforward to me). The ITN effect size overall seems much rosier in this model than in others, for example the model shown in Fig 9.9 of the 2022 World Malaria Report has effects on prevalence nearly gone by the end of 3 years for a realistic net campaign.

Thank you for your comment. Unfortunately, it is not trivial to change the net usage during a single simulation due to the deterministic nature of the model used and so this extension is beyond the scope of this work. We agree that caveats should be made more clearly in the results and have edited the first paragraph of the modelling results on L159-163 with this health warning:

“Results for these different types of protection are shown as their ability to reduce EIR in a hypothetical non-seasonal setting. The exact values should be treated with caution as they are reliant on uncertain model assumptions, though the relative magnitude of the different types of protection are likely to be more robust as they are less sensitive to these assumptions.”

Our net parameters have been taken from previously published models e.g. references 40, 41 and 63 which were able to recreate the results of cluster randomised control trials (reference 77), affording us some confidence that the results are broadly reliable.

Since indirect effects of nets have already been approximated both with models and with control/spillover trial arms, I’m not sure what new knowledge this paper adds. The authors could have used an individual-based model and tried to understand what role heterogeneity and inconsistent usage might play in modulating the effects of LLINs, which are otherwise predicted to be very strong, perhaps much stronger than have been observed in programmatic implementation. That kind of model could be really helpful for programs and the global community to understand why nets have not had more of an impact. What is the second part of the analysis that is not achievable with an individual-based model (from authors’ rebuttal)?

We believe this paper adds to the literature by carefully disentangling the indirect and direct components of protection to both users and non-users at different EIRs and levels of resistance that have not been presented in previous trials nor can be calculated now from trials due to understandable ethical reasons. We do not believe we would be able to tease apart these components in the same way with an individual based model. The paper is already complicated enough and risks becoming overly lengthy. We very much agree with the reviewer about the interest of the work they suggest but feel that this is outside the scope of this current paper.

Minor comments:

L61-2: don't net users also receive indirect protection?

Yes. This has been made clearer in L38-40:

“As shown in previous research such as Hawley et al. [13] and Killeen et al. [14, 15], LLINs offer different benefits for users and non-users within a community: net-users receive personal protection while both users and non-users benefit from indirect protection.”

L146-7: please put the peak difference here, otherwise it seems like you're trying to hide it

We have not included the peak difference because we believe this could be misleading when not taken into context. The peak difference (prevalence non-users – prevalence of users) is 1 in a cluster where all net users do not have malaria and all non-net users do. There are two clusters with very high usage > 85% where the children who have bed nets have malaria but the children who do not sleep under bed nets do not have malaria. This results in a difference of -1, but this is misleading because many more children are in the net user category than non-net category. We have instead added a sentence of L129-130 to explain this:

“We note that in some extreme clusters all users or non-users may have malaria so individual clusters may have a difference in prevalence between -1 and 1”.

Fig 1C: please add to the caption that the circle fill color is the difference in DHS. At first glance it's hard to see that there is any color at all and I was confused by L152-3 saying that Fig 1C shows cluster-level data. Please also clarify that net usage is all-age net usage, not U5 net usage (I think this is the case from reading the Methods)

Thanks. We have amended the caption for Figure 1 on L885-887 to say:

“The absolute difference in prevalence between LLIN users and non-users aged 6-59-months. The coloured tiles show the difference for our model estimates and the coloured points show the difference from the DHS survey (size represents number of data points).”

Since DHS data is 6-59 months, we present 6-59 month model estimates in Fig 1c so the comparison is fair. Figures 1a and 1b show all age prevalence as stated in the caption:

“Figure 1: Protected impact on malaria prevalence of standard pyrethroid long-lasting insecticidal nets (LLINs) for net users and non-users. (a) All-age malaria slide prevalence for a perennial setting with an initial entomological inoculation rate (EIR) of 100 bites per person per year. At year 1 (indicated by the vertical dashed black line), in this example, 50% of the

population switch to using LLINs. (b) The dependency of all-age malaria parasite prevalence on the annual initial EIR for the baseline pre-intervention scenario (black solid line), the LLIN users (blue dashed line), and non-users (red line). The initial EIR of 100, which is the simulation shown in A, is indicated by the vertical dashed line. (c) The absolute difference in prevalence between LLIN users and non-users aged 6-59-months. The coloured tiles show the difference for our model estimates and the coloured points show the difference from the DHS survey (size represents number of data points). (d) RDT prevalence from DHS surveys for users (blue) and non-users (red) for different cluster prevalences. 4138 clusters have been used for this figure (one value for users and non-users), with the centre of the box and whisker plots showing the median, outside lines showing the first and third quartiles, and the whiskers indicating 1.5 times the interquartile range. Dots indicate outliers.”

We have included the following line in the methods on L502-503”:

“Depending on the analysis both all-age prevalence and 6-59-month old prevalence are presented.”

L177-8: what is meant by “magnitude of the benefit provided by the barrier and the community effect”? from your results, the barrier can have a community effect, so this phrasing is confusing

Thank you, this was a typo. We have changed L155-156 to:

“The mechanistic model suggests that the relative magnitude of the direct and the community effect varies according to the level of disease endemicity and usage of nets.”

L179-82: I found these sentences confusing, in addition to a control there are 4 scenarios and a control (so a total of 1+4+1=6 scenarios)? But from Fig 2 it looks like there are a total of 5 scenarios?

Thanks for this. We only have one control. We have changed L157-159 to:

“In addition to a control where nobody in the community is given any nets, we identify 4 scenarios that describe the different types of protection offered by untreated nets or LLINs to users and non-users (Figure 2).”

L213-4: does the 11% EIR reduction to the community include non-users, whose EIR is fixed? Is it that their EIR would go down (by some amount not specified) if allowed to decrease? This is confusing.

Sorry, there was some confusion here caused by our wording. EIR is not fixed so it can change in both users and non-users (other than controls). There is no direct benefit on non-users as, by our definition, these people do not use a net so all benefits are indirect. This has been explained in detail in the response to the first comment above and the following sentence has been changed to make the wording clearer.

“We can also summarise the direct benefit of the barrier to the combined community (users and non-users). When one person in the community uses a net, the relative reduction in EIR at the community level is minimal (0% [95% UI: 0% - 1%]). However, when 80% of the community are using nets, there is a relative reduction in EIR of 11% [95% UI: 5% - 25%] to the community.”

L244-8: a little confusing to me how this is not rolling in some indirect benefit to users.

We have tried to better explain our differentiation between direct and indirect benefits. Please see our first response above.

L249: how is it the “same pattern”? it looks like the opposite pattern to me (more vs less benefit at high coverage)?

Thanks for noticing this. We have clarified this on L231-233 :

“The same pattern occurs at the community level with the direct benefit of insecticide as the direct benefit of the barrier with negligible relative reduction in EIR for one individual using a LLIN but 14% [95% UI: 5% - 26%] when 80% of people are using them.”

L252-3: isn't the effect on attempts also included in scenario a? how can there be even more attempts when there is insecticide? Is it due to the repellency effects?

Yes, this is correct – L233-235 now says

“This is again caused by the deterrent/repellent nature of the insecticide which dissuades mosquitoes from entering houses with LLINs and increases the chance they feed elsewhere.”

L279-82: I thought there was zero benefit to non-users when only 1 person uses a net? Perhaps rephrase this sentence (?)

There is zero benefit to non-users when 1 person uses a net but we are comparing non-users of a net at 80% usage to a single user.

“It is interesting to note that the model predicts ~10 % more protection is provided to people not sleeping under a net in a community with 80% LLIN use than would be provided by a single LLIN user in a community of non-users.”

L315-6: greater relative reductions in EIR seen at higher levels of resistance: this is the opposite of what Fig 4 shows, no?

Yes, this was a mistake – thanks for point this out. L299-301 now says:

“Overall, the relationship between the level of resistance and protection is non-linear, with greater relative reductions in EIR seen at lower levels of resistance in the local mosquito population (Figure 4, Supplementary Fig. 10).”

L350-3: sentence beginning “Similarly” is confusing. Very little difference in prevalence between users and non-users? A change in prevalence in each of the two groups, or a change in the difference in prevalence between the two groups?

Thanks – have edited the sentence on L336-339 to say:

“Similarly, our model predictions show very little difference between users and non-users in the presence of insecticide resistance and we could not see a change in prevalence of the two groups over time or with increasing pyrethroid resistance in local mosquitoes in the empirical data.”

L359-65: for some of these factors, haven't you already shown that you think some have little effect?

We haven't investigated the impact.

L372: "routine data" I would not at all consider DHS data to be routine data. Routine data usually refers to information that comes from surveillance systems (cases, interventions, etc), and DHS is survey data not from routine systems.

We agree on reflection and have changed "routine" to "survey". Thanks!

L377-8: does "seasonality" refer to seasonality of net use, or seasonality of transmission?

Thanks for pointing out, we have changed to "seasonality of transmission".

L384-5: why is a broad geographical and temporal period needed for a validation dataset? Why not a research dataset that includes a high level of heterogeneity in the research site?

This is a valid point. A broad geographical and temporal dataset would be needed to explore the impact of different LLIN usage and pyrethroid resistance in mosquitoes though isn't necessary for this analysis in principle. Data would be needed prior to the introduction of LLINs so we would have to rely on early cluster randomised control studies.

We have replaced the sentence,

"We are unaware of such a dataset which spans the broad geographical and temporal period available in the DHS data"

With on L372-374

"This could be possible through re-analysis of results from early cluster randomised control trials where heterogeneity could be quantified before and after the introduction of ITNs."

L380-1: are you saying exposure fluctuates annually like net use does (seasonally with vector abundance)? Or that there is variability in exposure across individuals living in the same area?

Both. L367-369 now says:

"Evidence suggests that net use fluctuates annually [60], and that exposure to infectious bites may be equally variable given differences in human activity within a community and mosquito seasonal activity [61]."

L417-8: are there any analyses of DHS data in the literature looking at net access within households and spatial variation in access that could inform on this?

We are unaware of analysis of this type and agree it would be useful to understand the mechanisms going on. However, we do not think there is sufficient temporal and spatial information in the majority of DHS dataset to do this and, as far as we are aware DHS does not record net types.

L435-7: the DHS data are what they are, are you saying that your model result looking at the DHS

data and pyrethroid resistance are especially uncertain? I think you are not interpreting the DHS data directly here, correct? Are your estimates of community resistance particularly “variable” or just very uncertain due to the nature of the resistance data and spatial scales you’re working with?

The pyrethroid resistance data is from a different source with coarser granularity and the resistance may not well match with the cluster locations. We have clarified this on L423-426:

“Care should be taken interpreting the DHS data combined with our estimates of the level of resistance in a community using the discriminating dose bioassay because the resistance estimates are highly variable and have been shown to be heterogeneous across the districts.”